# Miniature neurotransmission is required to maintain Drosophila synaptic structures during ageing

Soumya Banerjee ⬤ [1], Samuel Vernon[1], Wei Jiao[1], Ben Jiwon Choi[2], Evelyne Ruchti[1], Jamshid Asadzadeh ⬤ [1], Olivier Burri[1], R. Steven Stowers[3] & Brian D. McCabe ⬤ [1✉]

The decline of neuronal synapses is an established feature of ageing accompanied by the diminishment of neuronal function, and in the motor system at least, a reduction of behavioural capacity. Here, we have investigated *Drosophila* motor neuron synaptic terminals during ageing. We observed cumulative fragmentation of presynaptic structures accompanied by diminishment of both evoked and miniature neurotransmission occurring in tandem with reduced motor ability. Through discrete manipulation of each neurotransmission modality, we find that miniature but not evoked neurotransmission is required to maintain presynaptic architecture and that increasing miniature events can both preserve synaptic structures and prolong motor ability during ageing. Our results establish that miniature neurotransmission, formerly viewed as an epiphenomenon, is necessary for the long-term stability of synaptic connections.

[1] Brain Mind Institute, EPFL - Swiss Federal Institute of Technology Lausanne, Lausanne, Switzerland. [2] Department of Biology, New York University, New York, USA. [3] Department of Cell Biology and Neuroscience, Montana State University, Bozeman, USA. ✉email: brian.mccabe@epfl.ch

Diminishment of synaptic structures in the brain is a hallmark of ageing and correlated with the decline of many cognitive parameters including learning, memory and executive function[1–4]. Age-dependent alterations of synapses are also evident in the motor systems of both humans and rodents. Neuromuscular junction (NMJ) terminals have been described to shrink and partition with ageing, accompanied by reduced neurotransmission[5–12] and progressive diminution of motor ability[13,14]. However, establishing a causal relationship between age-dependent changes in synaptic structure, neuro-transmission properties and behavioural curtailment has proven challenging[1–4].

Here, using a system to investigate the ageing of motor synapses in *Drosophila*, we find that analogous to mammals, *Drosophila* synaptic terminals fragment as ageing progresses, accompanied by a decline of both neurotransmission and motor ability. Surprisingly, through experiments designed to causally relate these phenomena, we find that vesicular neurotransmission is necessary to maintain synaptic structures during ageing. Moreover, using neurogenetic tools that allow individual interrogation of each vesicular neurotransmitter release modality, we discover that miniature events are singularly essential to preserve synaptic structures and maintain motor ability as animals age. Our results reveal that miniature neurotransmission is uniquely critical for the long-term structural and functional conservation of synapses during ageing.

## Results

**Motor synaptic terminals fragment, neurotransmission is reduced and motor ability declines with ageing.** To investigate the structural stability of adult *Drosophila* synapses as they age, we examined a subset of glutamatergic NMJ terminals produced by motor neurons that express the transcription factor HB9[15]. These include a small subset of motor neurons that form bilateral terminals innervating the musculi ventralis interni mediales (mvim)[16] muscles in abdominal segment A2, but do not include the majority of motor neurons in this segment or these motor neurons in more posterior segments (Fig. 1a, b). Larval *Drosophila* abdominal muscles have tonic neuromuscular terminals which release only glutamate and also phasic terminals which release neuropeptides in addition to glutamate, each of which are produced by distinct neurons[17,18]. Similarly, we found adult mvim muscles had both tonic (mvim$^T$) and phasic (mvim$^P$) terminals which could be distinguished by vesicular monoamine transporter (VMAT) labelling of the phasic terminal (Supplementary Fig. 1a). However, only the tonic mvim$^T$ terminal expressed the HB9 transcription factor (Supplementary Fig. 1a). As is typical of *Drosophila* synapses[19], the mvim$^T$ presynaptic terminal is composed of a chain of linked bouton varicosities each of which contains multiple active zones for vesicle release (Fig. 1b, c). We examined the morphological characteristics of these terminals in young adult animals from 5 days after eclosion throughout lifespan (Fig. 1d–i; Supplementary Tables 1 and 2). During this period postsynaptic muscle surface area did not significantly change at any timepoint (Supplementary Fig. 1b). In contrast, presynaptic terminal area declined with ageing ($R^2 = 0.13$, $p < 0.001$) (Supplementary Fig. 1c) accompanied by an obvious progressive decline in synaptic bouton diameter ($R^2 = 0.28$, $p < 0.001$) such that the average bouton diameter was 74% ($p < 0.001$) smaller in old animals compared to young animals (Fig. 1j–p). This morphological size reduction was accompanied by a change in the distribution of active zone synaptic vesicle release sites labelled by the marker Bruchpilot (Brp)[20] (Fig. 1j–o), which increased in number until middle age before declining in older animals, consistent with prior observations in the central nervous system

(CNS) (Fig. 1q)[21]. Strikingly, boutons with only a single active zone which accounted for only 9% of the varicosities in young animals, steadily increased to 31% ($P < 0.001$) of all boutons in the oldest animals ($R^2 = 0.48$, $p < 0.001$) (Fig. 1r and Supplementary Table 2). This observation was consistent with larger multiple active zone synaptic boutons cleaving or fragmenting into smaller units with less active zones as animals aged, in the most extreme case being reduced to having only a single active zone. Consistent with this conclusion, the total number of morphologically distinct albeit smaller boutons increased steadily with ageing (Supplementary Fig. 1d) even as total terminal area declined (Supplementary Fig. 1c). Moreover, in young animals, we observed that individual presynaptic boutons were completely enveloped by a matching single postsynaptic membrane elaboration (Fig. 1s). In contrast, in old animals, multiple smaller presynaptic boutons were surrounded by a single postsynaptic elaboration with interstitial gaps occupied by peripheral glia (Fig. 1s, t). This observation was further supported by serial electron microscopy of young and old synaptic terminals (Fig. 1u, v). Consistent with our light microscope observations, we observed the frequent presence of small synaptic boutons containing an active zone in the terminals of old animals connected by thin processes to other boutons within the terminal (Fig. 1v), in marked contrast to the large boutons of young animals (Fig. 1u). The connections between small boutons in old animals coupled with the presence of multiple presynaptic and postsynaptic proteins similar to those found in larger young boutons (Supplementary Fig. 1e), are consistent with these small boutons remaining functional. To determine if bouton fragmentation was a general feature of ageing of *Drosophila* terminals, we also examined boutons of synapses innervating proximal femur leg muscles. Similar to abdominal synapses, we observed age-dependent fragmentation of boutons at these terminals (Supplementary Fig. 1f) consistent with fracturing of boutons into smaller units being a universal feature of *Drosophila* NMJ synaptic terminal ageing.

Having established a pattern of synaptic structural fragmentation as *Drosophila* terminals aged, we next examined the functional properties of these terminals. Two primary forms of neurotransmission occur at fast chemical synapses—evoked neurotransmission where action potentials trigger the release of multiple synaptic vesicles and miniature neurotransmission (also known as spontaneous release) where single synaptic vesicles are released in the absence of an action potential[22,23]. When we examined the amplitudes of evoked neurotransmission at these terminals as ageing progressed, we observed an initial rapid decline in younger animals followed subsequently by a less precipitous but steady reduction of evoked amplitudes ($R^2 = 0.31$, $p = 0.004$) (Fig. 2a–c and Supplementary Table 2). We examined if these changes in neurotransmission were associated with alterations of presynaptic active zone size or Brp intensity (Supplementary Fig. 2a, b). We noted that active zone size significantly decreased with ageing while Brp intensity increased (Supplementary Fig. 2a, b), similar to prior observations in the CNS[21].

In contrast to evoked release, miniature excitatory junctional current (mEJC) amplitude remained constant throughout lifespan ($R^2 = 0.11$, $p = 0.04$) apart from the very oldest animals which also had a sharp decline of muscle input resistance (Fig. 2d and Supplementary Table 2). As expected, the progressive reduction of EJC amplitudes while mEJC amplitudes remained mostly constant was reflected in a significant decline ($R^2 = 0.23$, $p = 0.03$) of quantal content with ageing (Supplementary Fig. 2c). We also examined if postsynaptic glutamate receptor puncta size changed with ageing. We found that the size of postsynaptic receptor puncta did increase until middle age but remained elevated in older animals, confounding an obvious contribution

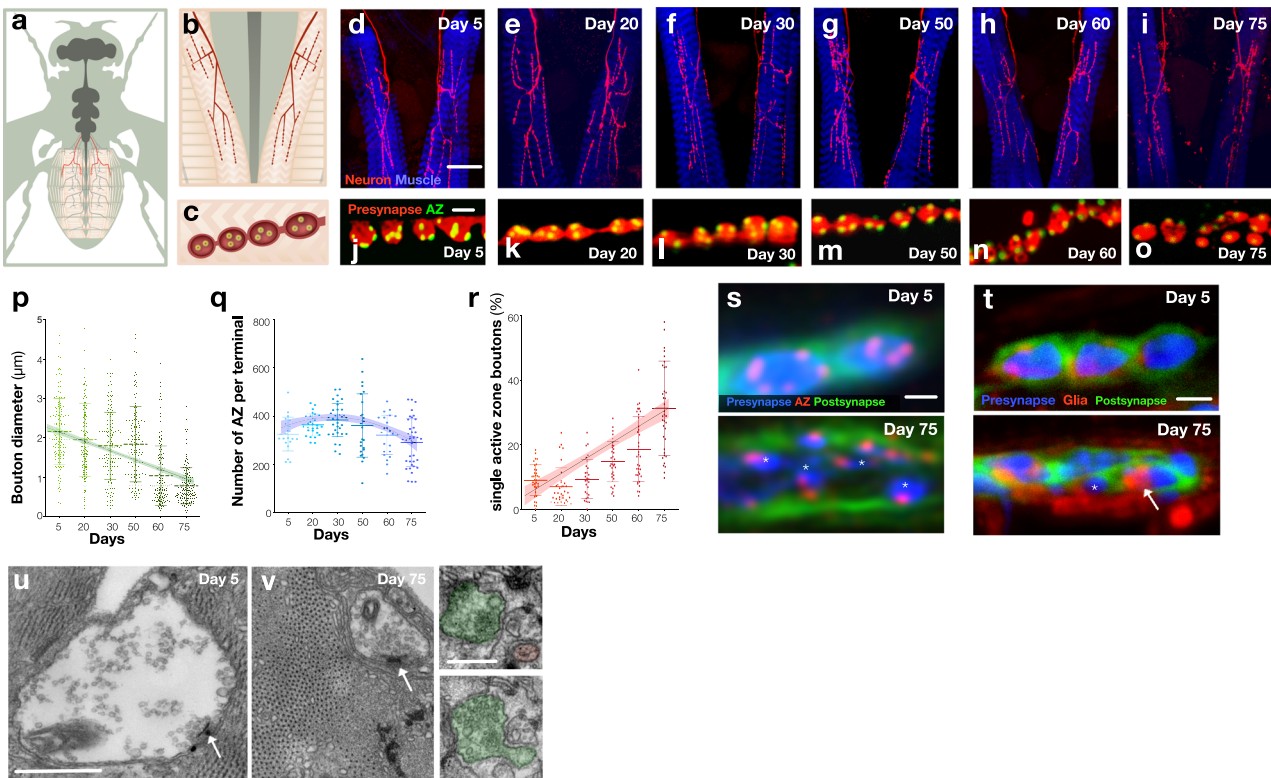

**Fig. 1 *Drosophila* synaptic terminal structures fragment, neurotransmission declines and motor behaviour is diminished during ageing.** Schematic of adult *Drosophila* HB9 labelled motor neurons (**a**, red) which innervate a subset of ventral and dorsal muscles of the anterior abdomen including A2 mvim (**b**) producing terminal bouton varicosities (**c**) with multiple active zones of synaptic vesicle release (yellow). **d–i** Representative images of progressive degeneration of HB9 abdominal synaptic terminals (red, GFP) innervating A2 abdominal muscles (blue, phalloidin) from 5 days after eclosion to 75 days. Magnification is identical for all images. **j–o** Representative images of progressive fragmentation of synaptic boutons (red, GFP) through reduction in size and number of active zones (green, Brp) per bouton during ageing. Magnification is identical for all images. **p** Quantification of diminishment of bouton diameters during ageing. **q** Quantification of the number of active zones during ageing. **r** Quantification of bouton fragmentation during ageing as measured by the percentage of boutons with only a single active zone. **s** Representative images of presynaptic boutons (blue, mCherry) at day 5 (upper panel) and day 75 (lower panel). Boutons are labeled with the active zone marker Brp (red) and the postsynaptic membrane marker Dlg (green). Asterisks indicate fragmentation of presynaptic terminal at day 75. Magnification is identical for both images. **t** Example of postsynaptic membrane (Green, Dlg) becoming opposed by peripheral glia (Red, mCherry) [indicated by arrow] as presynaptic terminals fragment (indicated by asterisk) (Blue, GFP) in 75-day old animals (lower panel), which is not found in 5-day-old young animals (upper panel). Magnification is identical for both images. $n = 4$ biologically independent preparations. **u**, **v** Representative micrographs of bouton at day 5 and day 75, respectively. **v** (Right panels) serial micrograph shows an example of a fragmented bouton which is associated with other boutons by a thin process. Arrows indicate T-bars. $n = 11$ (**u**) and 5 (**v**) biologically independent preparations. One-way ANOVA, ***$p < 0.001$, $R^2 = 0.28$, $n = 150$ boutons per timepoint (**p**), ***$p < 0.001$, $R^2 = 0.11$, $n \geq 21$ presynaptic terminals per timepoint (**q**), ***$p < 0.001$, $R^2 = 0.48$, $n \geq 27$ presynaptic terminals per timepoint (**r**), Tukey's multiple comparison tests were used to compare the mean of each timepoint with the mean of all other timepoints (ns = not significant, *$p \leq 0.033$, **$p \leq 0.002$, ***$p \leq 0.001$). All statistical analysis details with a precise value of '*n*' are reported in Supplementary Table 2 and in Source Data file. Scale bars, 40 μm (**d–i**), 2 μm (**j–o**, **s**, **t**), 500 nm (**u**, **v**), 250 nm (**v**, right panel). Data are represented as mean ± SD. $n =$ biologically independent samples.

of this alteration to age-dependent neurotransmission changes (Supplementary Fig. 2d). To further investigate the age-dependent decline of neurotransmission, we measured the readily releasable pool (RRP) of synaptic vesicles at this terminal in young and old animals. We found a 56% ($p < 0.001$) reduction of RRP pool size between young and old animals (Supplementary Fig. 2e), consistent with observations of reduced evoked amplitudes and diminished quantal content with ageing. In contrast to miniature event amplitudes, miniature event frequency however did decline steadily as animals aged ($R^2 = 0.16$, $p = 0.004$), much like evoked amplitudes (Fig. 2e). These results suggested primarily presynaptic changes in neurotransmission during ageing, with a progressive decline of both evoked neurotransmission and miniature event frequency as ageing advances.

In tandem with these changes in neurotransmission, motor ability, measured by negative geotaxis climbing, also steadily declined as animals aged, consistent with previous reports[24] ($R^2 = 0.86$, $p < 0.001$) (Fig. 2f). We also examined synaptic terminal morphology and functional properties when ageing was accelerated or decelerated by changing the temperature at which the animals were cultured (Supplementary Fig. 3a–v, Supplementary Table 3). We observed a conserved relationship between lifespan and bouton fragmentation, with the frequency of fragmented bouton accumulation correlating with relative lifespan stage (Supplementary Fig. 3q, v, Supplementary Table 3). In sum, concomitant with a diminishment of motor ability as *Drosophila* age, both motor neuron evoked amplitudes and miniature event frequency decline accompanied by fragmentation of synaptic structures (Fig. 2g).

**Inhibition of vesicular neurotransmission induces premature fragmentation of synapses.** To begin to dissect the causal relationship between the synapse ageing phenomena we observed, we

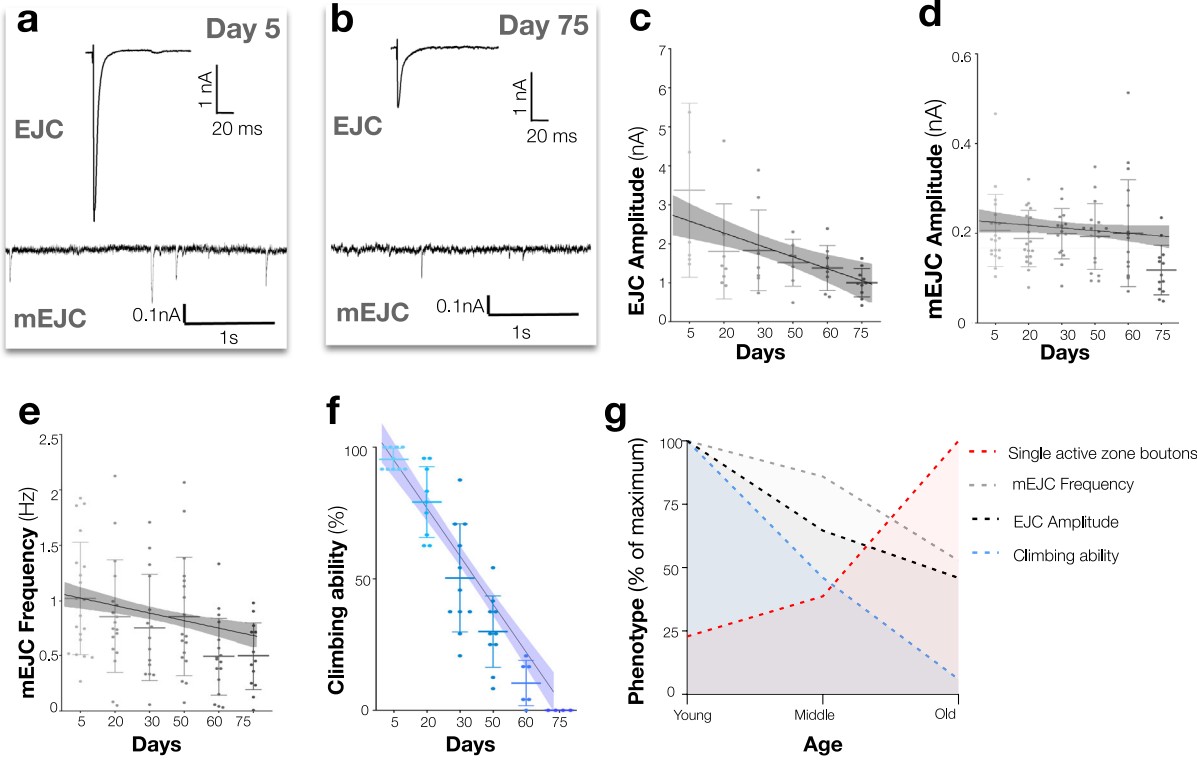

**Fig. 2 Neurotransmission declines and motor behaviour is diminished during ageing. a, b** Representative traces of excitatory junctional currents (EJCs) (above) and miniature excitatory junctional currents (mEJCs) (below) from 5-day-old and 75-day-old A2 mvim terminals. **c** Quantification of EJC amplitude decline during ageing. **d** Quantification of mEJC amplitude stability during ageing. **e** Quantification of mEJC frequency decline during ageing. **f** Quantification of declining motor ability during ageing. **g** Summary diagram indicates the progressive change of key ageing parameters as a percentage of the maximum observed phenotype. Compared to young animals (day 5) individual timepoints were significantly different at middle age (day 30) for increased bouton fragmentation, early old age (day 60) for EJC amplitude, early old age (day 60) for mEJC frequency and middle age (day 30) for motor ability. One-way ANOVA, **$p = 0.004$, $R^2 = 0.31$, $n \geq 7$ presynaptic terminals per timepoint (**c**), *$p = 0.04$, $R^2 = 0.11$, $n \geq 13$ presynaptic terminals per timepoint (**d**), **$p = 0.004$, $R^2 = 0.15$, $n \geq 13$ presynaptic terminals per timepoint (**e**), ***$p < 0.001$, $R^2 = 0.86$, $n \geq 5$ groups per timepoint (**f**), Tukey's multiple comparison tests were used to compare the mean of each timepoint with the mean of all other timepoints (ns = not significant, *$p \leq 0.033$, **$p \leq 0.002$, ***$p \leq 0.001$). All statistical analyses with a precise value of '$n$' are reported in Supplementary Table 2 and in Source Data file. Data are represented as mean ± SD. $n =$ biologically independent samples.

wished to examine the effects upon the synaptic structure of depreciating neurotransmission. To do this without compromising overall animal behaviour or lifespan, we manipulated only the restricted HB9-expressing subset of neurons during ageing. Moreover, as age-dependent synaptic structural changes are similar across culturing temperatures (Supplementary Fig. 3), we employed temperature selective gene expression[25] to restrict these manipulations only to mature adults. We first generated a genome engineered conditional protein null allele of the sole *Drosophila* vesicular glutamate transporter (VGLUT) to selectively deplete vesicular neurotransmission in adult HB9 neurons and confirmed efficacy by immunohistochemistry (Supplementary Fig. 4a, b). As expected[26], removal of *vglut* strongly reduced both evoked amplitude and miniature frequency (Fig. 3a and Supplementary Table 4) in HB9 neurons while the amplitude of the few remaining miniature events was unperturbed (Fig. 3a). When we examined the structure of these VGLUT depleted synapses, we observed no change in terminal morphology in the first few days of adult life (Fig. 3b; Supplementary Fig. 4c, d; Supplementary Table 5). However, subsequently, we observed a rapid decline in bouton size (Fig. 3b) and accumulation of fragmented single active zone boutons, accompanied by a reduction in the presynaptic terminal area and the total number of active zones (Fig. 3b, Supplementary Fig. 4c, d). This process accelerated such that by 20 days we observed a 49% ($p < 0.001$) reduction in bouton size, and 81% ($p < 0.001$) more fragmented boutons than

controls in these young animals (Fig. 3b). Fragmentation of these terminals was not further enhanced by additional ageing suggesting a ceiling for the process (Fig. 3b). As a control, terminals of neurons within the same animals that do not express HB9, and thus were not manipulated, were unaffected (Supplementary Fig. 4e, f). This result established that a depreciation of either or both evoked and miniature neurotransmission in young animals was sufficient to induce bouton size reduction and fragmentation reminiscent to that observed in old animals. This suggested that a decline in neurotransmission could trigger synaptic bouton structural dissolution.

**Miniature, but not evoked, neurotransmission is required to maintain synaptic structures.** The majority of neurotransmitter released at *Drosophila* motor neuron terminals is via evoked release[27], so we next depleted evoked neurotransmission alone by inhibiting the voltage-dependent sodium channel Para[28] in adult HB9 neurons. As predicted, this abolished evoked release (Fig. 3c and Supplementary Fig. 4i) but miniature event amplitude and frequency were unchanged (Fig. 3c, Supplementary Fig. 4i and Supplementary Table 4). When we examined the structure of terminals where evoked neurotransmission was absent, we observed no changes in bouton size or morphology, including no increase in bouton fragmentation at any timepoint during ageing (Fig. 3d, Supplementary Fig. 4g, h; Supplementary Table 5). We

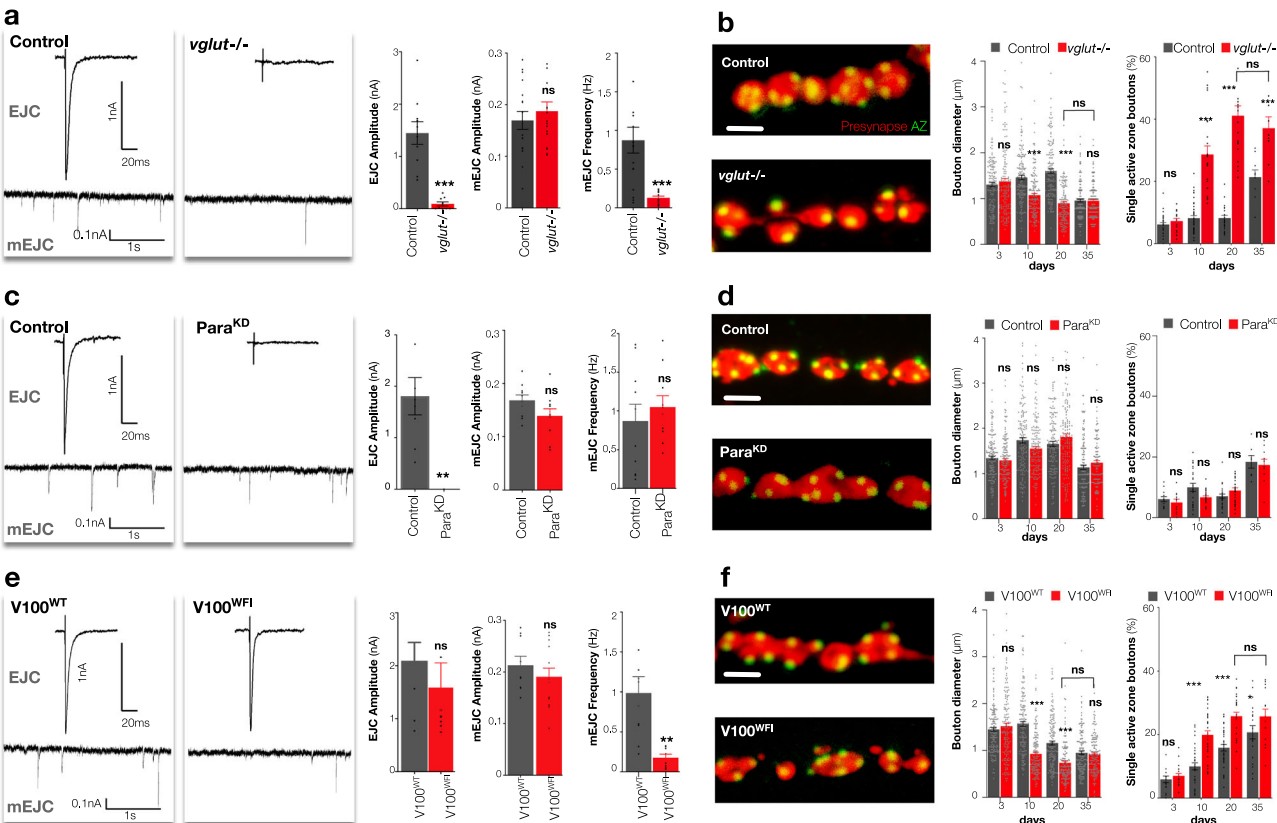

**Fig. 3 Miniature but not evoked neurotransmission is required for adult synapse maintenance. a** Representative EJC (above) and mEJC (below) traces, quantification of EJC amplitude, mEJC amplitude and frequency of control [UAS > GFP/+; HB9 > Gal4, Tub > Gal80ts] and HB9 neuron selective adult conditional vglut$^{-/-}$ mutants [Df(2R)371/B3RT_vglut_B3RT; HB9 > Gal4, UAS > GFP, Tub > Gal80ts/UAS > B3] at 20 days after eclosion. **b** Representative images of presynaptic HB9 neuron boutons (red, GFP) and active zones (Brp, green), quantification of bouton diameters and percentage of boutons with only a single active zone (bouton fragmentation) of control and vglut mutants until 35 days after eclosion. **c** Representative EJC and mEJC traces, quantification of EJC amplitude, mEJC amplitude and frequency of control [UAS > luciferaseRNAi/HB9 > Gal4, UAS > GFP, Tub > Gal80ts] and HB9 neuron selective adult conditional Para knockdown mutants (ParaKD) [UAS > GFP/+; HB9 > Gal4, Tub > Gal80ts/UAS > ParaRNAi] at 20 days after eclosion. **d** Representative images of presynaptic HB9 neuron boutons and active zones, quantification of bouton diameters and bouton fragmentation as measured by the percentage of boutons with only a single active zone of control and ParaKD mutants until 35 days after eclosion. **e** Representative EJC and mEJC traces, quantification of EJC amplitude, mEJC amplitude and frequency of HB9 neuron selective adult conditional V100WT [UAS > V100³'UTR_RNAi/UAS > V100WT; HB9 > Gal4, UAS > GFP, TubGal80ts] and V100WFI mutants [UAS > V100³'UTR_RNAi/UAS > V100WFI; HB9 > Gal4, UAS > GFP, TubGal80ts] at 20 days after eclosion. **f** Representative images of presynaptic HB9 neuron boutons and active zones, quantification of bouton diameters and percentage of boutons with only a single active zone of V100WT and V100WFI mutants until 35 days after eclosion. Unpaired two-tailed t-test, ***p < 0.001, n ≥ 9 presynaptic terminals per timepoint (**a**, EJC amplitude), p = 0.249, n ≥ 17 presynaptic terminals per timepoint (**a**, mEJC amplitude), ***p < 0.001, n ≥ 17 presynaptic terminals per timepoint (**a**, mEJC frequency), ***p < 0.001, n ≥ 6 presynaptic terminals per timepoint (**c**, EJC amplitude), p = 0.09, n ≥ 10 presynaptic terminals per timepoint (**c**, mEJC amplitude), p = 0.5, n ≥ 10 presynaptic terminals per timepoint (**c**, mEJC frequency), p = 0.4, n ≥ 8 presynaptic terminals per timepoint (**e**, EJC amplitude), p = 0.06, n ≥ 8 presynaptic terminals per timepoint (**e**, mEJC amplitude), **p = 0.004, n ≥ 8 presynaptic terminals per timepoint (**e**, mEJC frequency). Two-way ANOVA, followed by Sidak's multiple comparison tests were used to compare the mean of control and experimental genotypes in that timepoint, ns = not significant, ***p ≤ 0.001, n = 150 boutons per timepoint (**b**, bouton diameters), ns not significant, ***p < 0.001, n ≥ 9 presynaptic terminals per timepoint (**b**, single active zone boutons), ns not significant, n = 150 boutons per timepoint (**d**, bouton diameters), ns not significant, n ≥ 9 presynaptic terminals per timepoint (**d**, single active zone boutons), ns = not significant, ***p ≤ 0.001, n = 150 boutons per timepoint (**f**, bouton diameters), ns not significant, ***p < 0.001, n ≥ 14 presynaptic terminals per timepoint (**f**, single active zone boutons). All statistical analysis details with a precise value of 'n' are reported in Supplementary Tables 4 and 5 and in Source Data file. Scale bar = 2 μm (**b**, **d**, **f**). Experiments were carried out at 29 °C. Data are represented as mean ± SEM. n = biologically independent samples.

confirmed this result by expressing the inward rectifying channel Kir2.1[29,30] in HB9 neurons and found this also did not alter bouton morphology (Supplementary Fig. 4j, k). These results were in stark contrast to when both evoked and miniature neurotransmission were reduced in vglut mutants, suggesting that loss of evoked release was not the critical component of VGLUT depletion that elicited early presynaptic terminal fragmentation. Therefore, we next sought to inhibit the production of miniature events.

Unlike specific inhibition of evoked release, for which many pharmacological and genetic tools exist[23], investigation of the specific functions of miniature neurotransmission has been hampered by the lack of reagents with which to selectively perturb only this neurotransmitter release process in the presynapse without also simultaneously altering evoked neurotransmission[31]. However, recent amino acid level interrogation of proteins involved in neurotransmitter vesicle release has identified point mutations that can selectively deplete

miniature events[32]. V0 subunits of v-ATPase, in addition to roles in acidification, can alter vesicle membrane fusion through interaction with SNARE proteins[32]. A triple point mutant (WFI) of the *Drosophila* neuronal V0 subunit a1 (known as V100) cannot interact with Calmodulin and as a consequence the SNARE proteins Syntaxin1A and SNAP-25[32]. Unlike complete loss-of-function of V100[32] (Supplementary Fig. 5a), V100$^{WFI}$ mutants do not have defects in endolysosomal acidification and protein sorting or indeed of evoked neurotransmitter release, but rather have a specific disruption in the frequency of miniature events. To exploit this observation in adult neurons, we utilised an RNAi construct targeting the 3′UTR of the *V100* gene to selectively knockdown endogenous V100 protein in adult HB9 neurons, combined with transgenic restoration of either wild type V100 (V100$^{WT}$) or V100$^{WFI}$ mutant protein (Supplementary Fig. 5b). We found that evoked neurotransmission was not significantly altered in V100$^{WFI}$ rescued neurons compared to V100$^{WT}$ rescued controls (Fig. 3e and Supplementary Table 4). In contrast however, the frequency of miniature events was reduced by 83% ($p = 0.004$) in V100$^{WFI}$ mutants compared to V100$^{WT}$ controls (Fig. 3e), allowing us to examine the effects of a specific disruption of miniature neurotransmission on terminal morphology. When we examined the structure of V100$^{WFI}$ mutant terminals, we observed no initial change in bouton size or fragmentation (Fig. 3f and Supplementary Table 5), however by 10 days bouton sizes were reduced and bouton fragmentation increased, such that by 20 days, 40% ($p < 0.001$) more boutons were fragmented than controls (Fig. 3f). The total number of active zones and the presynaptic terminal area in V100$^{WFI}$ vs. V100$^{WT}$ expressing neurons were also significantly reduced (Supplementary Fig. 5c, d; Supplementary Table 5). Control unmanipulated terminals within the same animals were unaffected (Supplementary Fig. 5e, f). To determine if the morphological decline in V100$^{WFI}$ animals we observed was dependent only upon the depletion of miniature neurotransmission, we also examined V100$^{WFI}$ animals where evoked release was additionally abolished by simultaneous inhibition of Para (Supplementary Fig. 5g). We observed no change in bouton size, active zone number or the number of fragmented boutons in V100$^{WFI}$ mutants when evoked release was additional depleted compared V100$^{WFI}$ mutants alone (Supplementary Fig. 5h). This result supports that the structural dissolution in V100$^{WFI}$ mutants is an exclusive consequence of the diminished miniature neurotransmission in these animals. We also examined if the effects we observed were associated with presynaptic homeostatic plasticity (PHP), a process where presynaptic quantal content is increased in response to alterations of postsynaptic glutamate receptors[33]. As expected, we found presynaptic quantal content was increased in adult dGluRIIA mutants (Supplementary Fig. 6a)[34], however we observed no change in bouton size, active zone number or the number of fragmented boutons in dGluRIIA mutants compared to controls (Supplementary Fig. 6b), indicating PHP acts independently to the effects of miniature neurotransmission on terminal morphology. In sum, the premature accumulation of fragmented boutons when miniature neurotransmission was inhibited either as a component of VGLUT depletion or selectively in V100$^{WFI}$ mutants is consistent with a requirement for miniature neurotransmission and not evoked release being necessary to maintain presynaptic bouton structures.

**Increasing miniature events can retard age-dependent synapse fragmentation and preserve motor ability**. We speculated the effects on bouton morphology and fragmentation we had observed when miniature neurotransmission was depleted in young animals were mechanistically related to the size reduction

and fragmentation of boutons we had observed in older animals, where miniature events also decline. We therefore next sought to increase miniature events in aged animals. Complexin (Cpx) proteins bind to neuronal SNARE complexes and through helical domains can clamp neurotransmitter release[35]. Loss-of-function of Cpx induces a large increase in the frequency of miniature events but also reduces the amplitude of evoked release[36] (Supplementary Fig. 7a, b and Supplementary Table 4). The addition of amino acids between central and accessory helices of murine Cpx in a 'Helix Breaker' mutant (mCpx$^{HB}$) can restore evoked neurotransmission in Cpx mutant rescue experiments but does not rescue the aberrant increase in miniature events[37]. We utilised this observation in adult HB9 neurons to increase miniature events without perturbing evoked release by inhibiting *Drosophila* Cpx and co-expressing either wild-type murine Cpx (mCpx$^{WT}$) or a mCpx$^{HB}$ mutant (Supplementary Fig. 7c). Consistent with previous results[37], we found that the amplitude of evoked release was not significantly different when *Drosophila cpx* mutants were rescued with either mCpx$^{WT}$ or mCpx$^{HB}$ (Fig. 4a). In contrast, miniature events were increased by approximately twofold ($p < 0.005$) in mCpx$^{HB}$ rescued neurons compared with those rescued by mCpx$^{WT}$ (Fig. 4a and Supplementary Table 4) and this increase continued throughout lifespan (Supplementary Table 7). This manipulation allowed us to assay the effects of specifically increasing miniature neurotransmission as animals aged. When we compared the structure of synaptic terminals of mCpx$^{WT}$ and mCpx$^{HB}$ animals as ageing progressed, we found no differences between these conditions in young animals (Fig. 4b, Supplementary Fig. 7d, e). However, by 20 days we found a significant 15% ($p = 0.009$) increase in the average size of synaptic boutons and a 26% ($p > 0.001$) decrease in the number of fragmented boutons in mCpx$^{HB}$ vs. mCpx$^{WT}$ expressing animals (Fig. 4b; Supplementary Fig. 7d, e and Supplementary Table 6). This effect continued throughout ageing such that in the oldest animals, mCpx$^{HB}$ expressing terminals had 53% ($p < 0.001$) larger synaptic boutons and 39% ($p < 0.001$) fewer fragmented boutons than mCpx$^{WT}$ expressing animals (Fig. 4b and Supplementary Table 6). To determine if this retardation of age-dependent structural degradation was dependent only upon the increase of miniature neurotransmission, we depleted evoked release by inhibition of Para while simultaneously increasing miniature events by manipulating Cpx in old animals (Supplementary Fig. 7b). We found that eliminating evoked neurotransmission did not alter the benefits of increasing miniature neurotransmission upon age-dependent alterations of active zone number or bouton fragmentation (Supplementary Fig. 7f). Thus, specifically increasing the frequency of miniature events can reduce bouton fragmentation and preserve synaptic structures as animals age. Our results are consistent with the age-dependent decline of miniature neurotransmission inducing instability of synaptic terminal structures in old animals.

As increasing miniature events could preserve synaptic terminal structures in the HB9 subset of motor neurons, we next asked if increasing miniature neurotransmission in all motor neurons could alter the age-dependent decline of motor ability. To do this, we inhibited *Drosophila* Cpx in all adult motor neurons and rescued with either mCpx$^{WT}$ or mCpx$^{HB}$. We then measured motor performance as assayed by climbing ability. Similar to effects on synaptic structures, we found no differences in climbing ability when we compared these animals when very young. However, by 20 days, as the motor ability in mCpx$^{WT}$ expressing animals began to decline, mCpx$^{HB}$ expressing animals' motor ability was preserved to a level similar to young animals (Fig. 4c). This difference became more exaggerated during the middle of lifespan, where mCpx$^{HB}$ expressing animals had 49% ($p > 0.001$) better climbing ability than mCpx$^{WT}$ expressing

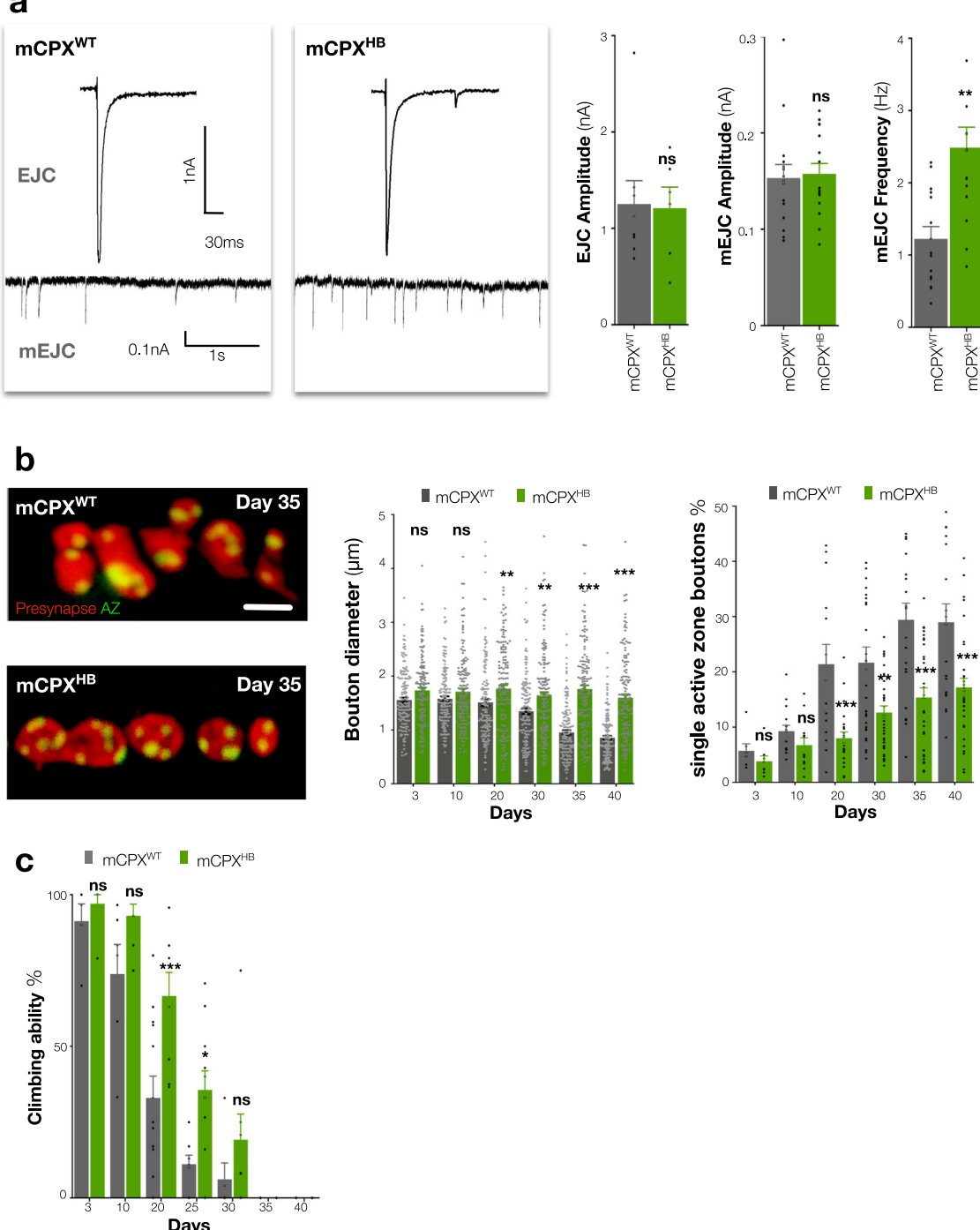

**Fig. 4 Increasing miniature neurotransmission preserves synapses and prolongs motor ability during ageing. a** Representative EJC (above) and mEJC (below) traces, quantification of EJC amplitude, mEJC amplitude and frequency of HB9 neuron selective adult conditional murine Cpx rescued wild type (mCpx^WT) [UAS > Cpx^RNAi/UAS > GFP; HB9 > Gal4, TubGal80^ts/cpx^SH1, UAS > mCpx^WT] and mCpx 'Helix Breaker' (mCpx^HB) [UAS > Cpx^RNAi/UAS > GFP; HB9 > Gal4, Tub > Gal80^ts/cpx^SH1, UAS > mCpx^HB] mutants at 20 days after eclosion. **b** Representative images of presynaptic HB9 neuron boutons (red, GFP) and active zones (Brp, green), quantification of bouton diameters and percentage of boutons with only a single active zone (bouton fragmentation) of mCpx^WT and mCpx^HB mutants until 40 days after eclosion. Images at same magnification. **c** Quantification of the climbing ability of motor neuron selective adult conditional mCpx wild-type mCpx^WT [UAS > Cpx^RNAi/OK6 > Gal4, UAS > GFP; Tub>Gal80^ts/cpx^SH1, UAS > mCpx^WT] and mCpx^HB [UAS > Cpx^RNAi/OK6 > Gal4, UAS > GFP; Tub > Gal80^ts/cpx^SH1, UAS > mCpx^HB]. Unpaired two-tailed t-test, $p = 0.89$, $n \geq 6$ presynaptic terminals per timepoint (**a**, EJC amplitude), $p = 0.81$, $n \geq 15$ presynaptic terminals per timepoint (**a**, mEJC amplitude), ***$p < 0.003$, $n \geq 15$ presynaptic terminals per timepoint (**a**, mEJC frequency). Two-way ANOVA, followed by Sidak's multiple comparison tests were used to compare the mean of control and experimental genotypes in that timepoint, ns not significant, ***$p \leq 0.001$, $n \geq 149$ boutons per timepoint (**b**, bouton diameters), ns not significant, **$p < 0.002$, ***$p < 0.001$, $n \geq 7$ presynaptic terminals per timepoint (**b**, single active zone boutons), ns not significant, *$p < 0.033$, ***$p < 0.001$, $n \geq 4$ groups per timepoint (**c**). All statistical analysis details with a precise value of 'n' are reported in Supplementary Tables 4 and 6 and in Source Data file. Scale bar = 2 μm (**b**). Experiments were carried out at 29 °C. Data are represented as mean ± SEM. $n$ = biologically independent samples.

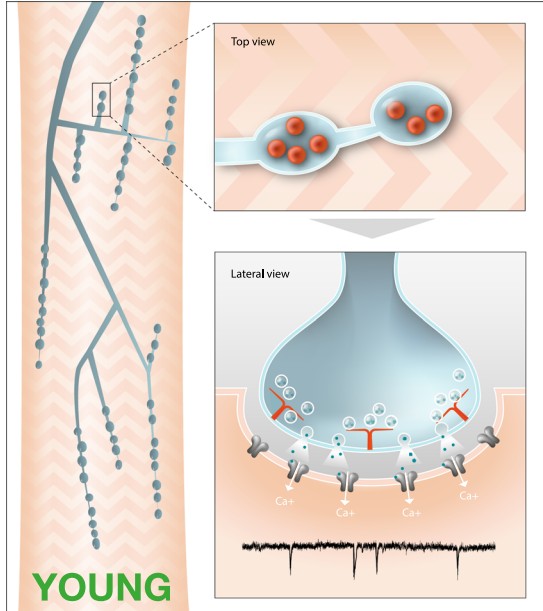
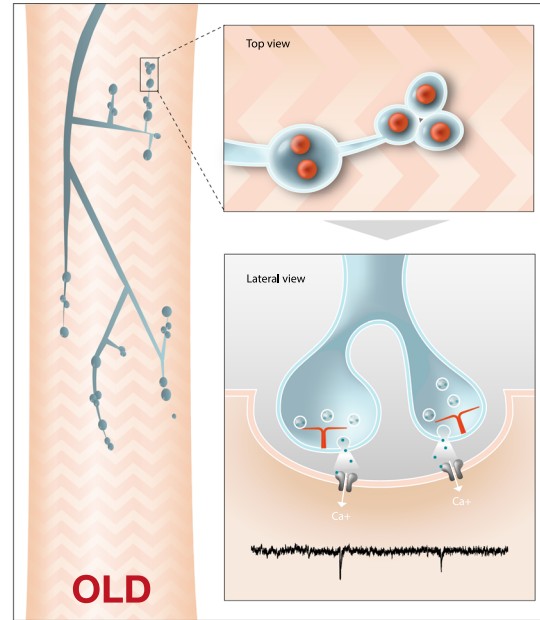

**Fig. 5 Model of NMJ synaptic structural changes in response to miniature event decline during ageing.** Miniature event frequency declines during ageing. In response, synaptic bouton diameters decrease and there is an increase in the number of small boutons with only a single active zone. All of these observations are consistent with young multiple active zone boutons fragmenting into smaller single active zone boutons as animals age. Reducing miniature events can accelerate this bouton fragmentation process in young animals while increasing miniature event frequency can delay synapse fragmentation in ageing animals.

animals (Fig. 4c). However, in the very oldest animals, where motor ability is profoundly disrupted, this significant difference was not maintained (Fig. 4c) and lifespan was not extended. These data establish that increasing miniature neurotransmission can not only maintain synaptic structures, but at least in middle-aged animals, also preserve motor ability.

## Discussion
Deterioration of both central and peripheral synaptic structures and alterations of neurotransmission is a conserved feature of ageing in both rodents and humans[4]. Consistently, using a preparation we have developed to examine adult synapses in *Drosophila*, we also observe age-dependent reductions of synaptic structure and neurotransmitter release. Interrogating this system, we find that age-dependent changes in vesicular neurotransmission, rather than being a consequence of alterations in synaptic structure, may instead precede and promote the decline of synaptic architecture. This finding is surprising and contrary to the generally assumed sequence of age-dependent synapse degeneration events[1–4]. Moreover, we establish that miniature neurotransmitter release is the key constituent of vesicular neurotransmission that is singularly required to maintain synaptic structures as they age (Fig. 5).

Since their discovery over 60 years ago[22], miniature events have been observed at every chemical synapse studied, but were often dismissed as 'noise' produced as a by-product of high fidelity evoked neurotransmission[38]. The supposition that miniature neurotransmission is an epiphenomenon has been maintained in part by the difficulty to discretely attenuate presynaptic miniature release in vivo without also simultaneously perturbing evoked neurotransmission. Here, by employing precise amino acid mutants of neurotransmission proteins derived from the intensive investigation of the mechanisms of vesicular release[23,39,40], in combination with adult-specific genetic manipulations, we reveal an essential and exclusive function for miniature events to maintain the structural integrity of synaptic terminals. We find

that reducing miniature events results in premature disintegration of synaptic terminals in young animals while increasing these events in older animals can not only preserve terminal morphology, but in addition retard the age-dependent decline of motor ability. Simultaneous inhibition of evoked release does not either further increase terminal degeneration when miniature neurotransmission is inhibited or limit the benefits to terminal structural integrity when miniature events are increased during ageing, indicating a unique and singular requirement for miniature events. We speculate that miniature release may be uniquely suited for processes necessary for the long-term stability of synapses as these events are produced continuously, in contrast to the stochastic and intermittent nature of evoked release. Our finding of the necessity of miniature neurotransmission for the long-term structural stability of synapses extends upon the ongoing re-evaluation of the functional importance of miniature events, in particular for synapse and circuit development[30,41,42].

While our data support a unique role for miniature neurotransmission in maintaining synaptic structural integrity, it remains to be determined how small amplitude miniature events can be discriminated from much larger evoked release to elicit their unique properties. In developing synaptic terminals, we have previously shown that the effects of postsynaptic depletion of miniature neurotransmission on presynaptic bouton maturation required the ionotropic activity of glutamate receptors[41]. Miniature events could potentially activate spatially distinct subpopulations of postsynaptic receptors to those activated by evoked neurotransmitter release[43,44] to trigger differentiating signalling cascades. In support of this, postsynaptic functional imaging with active zone resolution of *Drosophila* larval synaptic terminals has shown that the probability of synaptic vesicle release during evoked or miniature neurotransmission can be spatially segregated to distinct active zones[45,46]. Similar observations have been made in mammalian neurons[47]. Another possibility is that differences in the release kinetics between evoked and miniature neurotransmission could allow postsynaptic mechanisms to detect and discriminate miniature events.

For example, differential activation of Calmodulin can distinguish between local or global $Ca^{2+}$ signaling acting through voltage-gated $Ca^{2+}$ channels[48]. The ability to selectively inhibit either miniature or evoked neurotransmission in the presynapse, as described here, should enable further interrogation of these potential mechanisms.

We also describe here a *Drosophila* abdominal neuromuscular synapse preparation suitable to investigate mature adult synapses throughout their lifespan. We find that similar to intensively studied larval abdominal neuromuscular synapses[19,49], adult A2 mvim abdominal muscles also have both tonic and phasic innervation, though only the tonic terminal producing motor neurons expresses the HB9 transcription factor. The restriction of HB9 (and HB9 > Gal4) expression to a small subset of adult motor neurons, enables experiments, such as inhibition of neurotransmission throughout ageing as we have carried out here, without compromising overall animal behaviour or lifespan. We have also developed a voltage clamp electrophysiological approach for this muscle enabling direct measurement of synaptic currents and thus accurate quantification of neurotransmission properties, particularly important during ageing where potential confounding alterations of membrane potential and electrical resistance have been documented[9,50,51]. In this preparation, we have not observed an age-dependent increase in quantal content or 'set-point' as has been described at *Drosophila* proboscis NMJ terminals[50]. Rather we observe a progressive age-dependent decrease in evoked neurotransmission (in addition to miniature event frequency), quantal content and a reduction in the size of the ready realisable pool of synaptic vesicles. Our observations at this terminal are consistent with the age-dependent decline of neurotransmission described in other *Drosophila*, invertebrate and vertebrate synapses[9–12,52–54].

As we describe in *Drosophila* and as others have observed in mammals, miniature events decline as animals age[9,11]. Alterations of miniature neurotransmission have also been reported in the context of several neurodevelopmental, neurodegenerative and psychiatric diseases[55–58]. Our results suggest that, in addition to age-dependent synaptic decline, alterations of miniature events should be investigated further for a potential causal role in synaptic structural changes associated with brain disorders.

## Methods

**Drosophila stocks**. HB9 > Gal4 (subset of motor neurons)[15,59], OK6 > Gal4 (all motor neurons)[60], vglut>LexA (all motor neurons)[61], Df(2R)371 (deficiency covering *vglut*)[26], B3RT-Vglut-B3RT-LexA (this manuscript), UAS-Para$^{RNAi}$ (BDSC# 31676)[62], UAS-Kir$_{2.1E}$ (this manuscript), UAS > V100$^{3'UTR\_RNAi}$ (BDSC# 57860)[62], UAS > V100$^{WFI}$ [32], UAS > V100$^{WT}$ [32], UAS > Cpx$^{RNAi}$ [42], cpx$^{SH1}$ (Cpx null allele)[36], UAS > mCpx$^{WT}$ [37], UAS > mCpx$^{HB}$ (helix breaker)[37], BRP-GFP (BDSC#27786)[63], UAS > luciferase RNAi (BDSC#31603)[62], TubGal80$^{ts}$ [25], UAS > B3 (this manuscript), UAS > 6XGFP[64], UAS > 6XmCherry[64], LexOp>6XmCherry[64], Repo>QF2 (BDSC#66477)[65], QUAS > 6XmCherry[64], dGluRIIA$^{JA1}$ (this manuscript).

**Temporal control of targeted expression of transgenes in adult motor neurons**. The temporal control of targeted transgene expression was achieved by using the Gal4/Gal80$^{ts}$ system[25]. For the majority of experiments in this study where this technique was employed, crosses were raised at 18 °C and upon eclosion, male flies of the appropriate genotypes were separated and maintained at 18 °C for a further 3 days. They were then transferred to 29 °C to induce gene expression. An exception to this procedure was employed for conditional removal of *vglut* (Fig. 3a and Supplementary Fig. 4a–d), as VGLUT protein perjures for many days after *vglut* gene removal. In this case, animals were transferred to 29 °C 55 h after puparium formation, a period when adult abdominal NMJ synapses have completed development[15,66]. Consistent with no effects of this manipulation on synapse development, no differences were observed between *vglut* mutant terminals and controls at 3 days after eclosion (Fig. 3b and Supplementary Fig. 4c, d).

**Construction of conditional B3RT *vglut* null allele**. The B3RT_*vglut*_B3RT_LexA genome edit[67] involved Gibson assembly of a donor plasmid in which one B3 recombinase target (B3RT) was inserted into the 5'UTR of *vglut* and a second

B3RT in the *vglut* 3'UTR. The coding sequence of the LexA transcription factor was also included downstream of, and immediately adjacent to, the latter B3RT. Prior to B3 recombinase-mediated excision, *vglut* gene function is intact (i.e. viable over a *vglut* null allele) and LexA is not expressed. B3 recombinase-mediated excision results in: (1) a null allele of *vglut* due to the removal of the entire *vglut* coding sequence; and (2) a *vglut*>LexA reporter which was employed to confirm excision events (Supplementary Fig. 4b).

**Construction of UAS>B3**. B3 recombinase was amplified from pJFRC157-20XUAS-IVS-B3::PEST[68] (Addgene Plasmid #32136) using primers to add a syn21 translational enhancer sequence and remove the PEST domain. The resulting PCR product was transferred into pBID1_20xUAS, a derivative of the pBID expression vector[69] containing 20 UAS sequences.

*Construction of UAS>Kir$_{2.1E}$*. The enhanced murine Kir$_{2.1}$ mutant [E224G and Y242F][30] was PCR amplified from pCAG-Kir2.1-T2A-tdTomato (Addgene #60598). The synthetic Syn21 5'UTR sequence was added to enhance the expression level (Supplementary Table 8). Amplification was performed using Platinum Superfi polymerase (Invitrogen) with the primers described in Supplementary Table 8. The resulting PCR product was first transferred into pCR8GW-TOPO vector (Invitrogen) and then the resulting pCR8GW-Kir2.1E was transferred into pBID-UASC-G destination vector[69] to generate pBID-UAS>Kir2.1E. pBID-UAS>Kir2.1E was then inserted into the JMK22C (chromosome II) landing site using established methods (GenetiVision Services, Houston).

*Generation of dGluRIIA$^{JA1}$*. A novel GluRIIA mutant was generated via germline in vivo CRISPR-mediated gene editing. In short, animals expressing Cas9 in germline (BDSC # 67083) were crossed with individuals ubiquitously expressing an sgRNA (BDSC # 68059) that targets a genomic region within the GluRIIA coding sequence. Male progeny bearing all transgenes were recovered and crossed to a balancer line. Subsequently, individual stocks were made from single males in the next generation. Homozygote larva were then used to screen for lack of GluRIIA staining at larval NMJ terminals. Multiple mutant lines were recovered all of which were homozygous viable and fertile and confirmed to have no synaptic expression of dGluRIIA by immunohistochemistry. Three independent mutations were confirmed to introduce identical three nucleotide deletions (Drosophila genome nucleotides 5,556,441–5,556,443) predicted to remove Isoleucine 273 from the dGluRIIA ligand-binding domain, compromising either protein targeting or stability.

**Generation of anti-*Drosophila* VGLUT and VMAT antibodies**. *Drosophila* anti-VGLUT mouse monoclonal antibodies (clone 10D6G) were generated (Life Technologies Europe) using the C-terminal peptide sequence TQGQMPSYDPQ-GYQQQ of VGLUT coupled to KLH and validated by removal of staining in *vglut* mutants. In addition, a chicken anti-VGLUT antibody was raised against the same sequence (Life Technologies Europe) and validated similarly for experiments where monoclonal VGLUT antibodies could not be employed. *Drosophila* anti-VMAT (clone 3B5D12) mouse monoclonal antibody was generated (Eurogentec) using the N-terminal peptide sequence CTQGQMPSYDPQGYQQQ of VMAT[70] coupled to KLH and validated by immunohistochemistry.

**Generation of anti-*Drosophila* GluRIID antibody**. *Drosophila* anti-GluRIID Guinea Pig polyclonal antibody was generated (Eurogentech) using the C-terminal peptide sequence IESLKTDSEENMPVED of dGluRIID[71] coupled to KLH and validated by immunohistochemistry.

**Immunohistochemistry**. Adult flies were dissected in 1x PBS (Mediatech) along the dorsal midline to expose the ventral nerve cord (VNC) and abdominal muscles and were fixed for 40 min in 4% formaldehyde (Sigma-Aldrich) or for 6 min in Bouin's fixative (Sigma-Aldrich). Following fixation, preparations were washed multiple times in PBT (PBS + 0.3% Triton-X) and blocked in PBTB (PBT + 0.2% BSA) and PBTN (PBTB + 2% Normal Goat Serum). Primary antibodies used were Chicken or Rabbit anti-GFP (1:1000; AbCam), Rabbit anti-RFP (1:500; Clonetech), Mouse anti-DLG (1:100; Developmental Studies Hybridoma Bank [DSHB]), Mouse anti-Brp (Nc82, 1:100; DSHB), Rabbit anti-GluRIIC (1:100, Gift from Stephan Sigrist)[72], Mouse anti-GluRIIA (1:20; DSHB), Guinea Pig anti-GluRIID (1:500, this study), Mouse anti-VGLUT(1:100, this study), Chicken anti-VGLUT (1:100, this study), Rabbit anti-VGLUT (1:1000; Gift from Hermann Aberle), Mouse anti-VMAT(1:100, this study), Rabbit anti-Synaptotagmin (1:1000; Gift from Troy Littleton), Rabbit anti-Cpx (1:1000, Gift from Troy Littleton), Mouse anti-22C10 (1:100, DSHB), Mouse anti-tubulin (1:100, DSHB), Rabbit anti-Myc (1:500, Cell Signaling) and Guinea Pig anti-V100 (1:1000, Gift from Robin Heisinger). Preparations were incubated at 4 °C in primary antibodies and washed in PBTB and PBTN. Secondary antibodies were usually applied for 2–4 h at room temperature. Secondary antibodies (Life Technologies) used included goat anti-chicken (Alexa-488, 555 or 647; 1:400), goat anti-mouse (Alexa-Cy3 or 647; 1:400), goat anti-rabbit (Alexa-488, 546 or 647; 1:400) goat anti-Guinea Pig (555, 1:400), Alexa Flour 488-HRP (1:400; Jackson ImmunoResearch). To visualize body-wall muscles Phalloidin 647 (ThermoFisher, 1:200) was used. All immunostained

samples were mounted in ProLong Diamond antifade mountant (Invitrogen) and imaged in Zeiss 700 upright confocal microscope. Adult leg dissection, mounting and imaging were performed as described previously[73].

**Morphological analysis of muscles and synapses.** Dissected adult NMJs were labelled as described above. ImageJ including custom-designed ImageJ macros were employed to measure muscle area, synaptic terminal area, and number of active zones (available at https://doi.org/10.5281/zenodo.4912667)[80]. Analysis was carried out using Z stacks of Zeiss 700 confocal images of A2 mvim terminals. For internal controls, NMJ synapse morphology was quantified from presynaptic terminals, labelled by VGLUT, innervating A4 or A5 ventral muscles which do not express HB9 (Supplementary Figs. 4e, f and 5e, f). Average bouton diameter was determined by calculating the diameter of 150 boutons ($n \geq 5$ preparations) from the posterior region of presynaptic terminals innervating A2 ventral muscles. All morphological parameter data are provided in the Supplementary tables and source files. Original datasets are publicly accessible[80].

**Electron microscopy.** To intiate the preparation, 5- and 75-day-old adult flies were dissected in PBS, and then fixed in 3% glutaraldehyde and 0.5% paraformaldehyde in PBS (pH 7.2) overnight at 4 °C. The specimens were washed with PBS and post-fixed in 1% OsO4, dehydrated in alcohol, and embedded in Durcupan ACM (Fluka). Serial ultrathin sections were cut with a diamond knife (Diatome), collected on grids and stained with 1% uranyl acetate and lead citrate. The grids were then examined with a JEOL 1200 electron microscope. Images were quantified with ImageJ software. Original datasets are publicly accessible[80].

**Electrophysiology.** Dissections were performed at room temperature (20–22 °C) in HL3.1 saline (0 mM $Ca^{2+}$). Adult male flies were incapacitated on ice for 1–2 min prior to dissection. A single insect pin (0.1 mm stainless steel, Austerlitz, Slavkov, CZ) was placed through the back of the head into a 35 × 10 mm petridish filled with ~3 mm Sylgard (Dow-Corning, Michigan, USA). Legs and wings were removed with micro-dissection scissors (Fine Science Tools, California, USA). The most posterior tip of the abdomen was pinned and a dorso-longitudanal incision was made through the abdomen and thoracic musculature. Thoracic and abdominal hemisegments were pinned laterally, exposing A1–A6 muscle segments located along the ventral midline. Fat bodies and VNC were removed prior to recording. HL3.1 saline (1.5 mM $Ca^{2+}$)[74] was selected for recordings over AHL saline (1.5 mM $Ca^{2+}$) due to a significant reduction in muscle contractability during recordings. Recorded, synaptic currents were similar between AHL and HL3.1 salines. Discontinuous single electrode voltage clamp recordings[75–77] of adult neuromuscular glutamatergic synaptic currents were carried out using borosilicate glass electrodes (1B120F-4, World Precision Instruments, Florida, USA). Recording electrodes were pulled (Sutter P-1000, California, USA) to resistances of between 20-25 MΩ and filled with 3 M KCl. Suction electrodes (GC120T-10, Harvard Apparatus, Kent, UK) were fire polished to minimize damage to the nerve to a diameter of ~2–3 μm and filled with extracellular saline. EJCs were evoked (1 ms/1 Hz) with a 2100 isolated pulse stimulator (AM Systems, Washington, USA). Stimulus amplitude was set to 1.5–2 times the minimum threshold potential to ensure maximal axonal recruitment. mEJC and EJC recordings were made from individual muscle fibers of the A2 mvim abdominal muscle segments identified by muscle field position within the abdomen and through visualization of Hb9-Gal4 driven GFP positive synaptic terminals which directly innervate mvim A2 ventral muscles. Visualization of muscle field was conducted with an Olympus BX51-WI microscope (Olympus, Tokyo, JPN) together with a LUMPlan FI/IR 40x water immersion objective mounted on a TMC Micro-g air table (TMC, Massachusetts, USA) fitted with standard Faraday caging. Recordings were made using an Axoclamp 900 A amplifier (Molecular Devices, Sunnyvale, CA). Muscle fibers were held at −60 mV, recordings sampled at 20 kHz and lowpass filtered at 0.5 kHz using pClamp 11 (Molecular Devices, Sunnyvale, CA). Complete decay of the electrode potential between periods of current injection and voltage recording was visualized on a secondary oscilloscope during recording (Molecular Devices, Sunnyvale, CA), selected clamp switching frequency was 10 Khz, clamp gain was 1–2 nA/mV with current injection remaining at 30–40% of the duty cycle. Electrode capacitance neutralization was manually adjusted to reduce waveform decay and mitigate overshoot during recording. No correlation between injected current during voltage clamping and recorded synaptic current amplitudes were observed and voltage attenuation across the entirety of the muscle fibre was estimated to be at an isopotential to the clamp. Recordings were only accepted for analysis if muscle input resistance was >10 MΩ and injected current was <2.5 nA. mEJC amplitude and frequency were averaged over each 2-min recording using Mini Analysis 6.03 (synaptosoft). Evoked EJC amplitudes were averaged from 10 EJCs per recording. RRP was estimated as per[78] and adapted to suit the requirements of the adult NMJ. Briefly, recordings were conducted in 3 mM $Ca^{2+}$ extracellular saline and stimulated at 100 Hz. Cumulative amplitudes of evoked EJCs were plotted and a line of best fit was extrapolated back to 0 to give an estimate of overall cumulative pool size. This value was then divided by mean miniature amplitude in order to determine estimated RRP of each recording. Note, RRP measurements were made from one of six, electrically independent, muscle fibers. Thus, RRP estimates and for that matter, all physiological recordings, should be interpreted as

measurements for single muscle fibers rather than of the total muscle. During recording of HB9 > paraRNAi, we noted that the minimum current required to stimulate each EJC was at least twofold larger that of controls. We chose to utilise this measure as validation for the effectiveness of our RNAi construct. Though many factors may increase stimulation threshold, we rationalised this observation was consistent with the expected reduction of nerve conductance due to voltage-gated sodium channel depletion. Data presented in Fig. 3 and Supplementary Figs. 5–7 display failed EJCs resulting from stimulation at control levels. Original datasets are publicly accessible[79].

**Negative geotaxis motor ability assay.** For negative geotaxis climbing assays[24], 8–10 male flies of appropriate genotypes were sorted upon eclosion and transferred to new food vials every 2 days throughout the course of experiments. Groups of 8–10 male flies were transferred to a 22 cm long empty plastic vial containing no food with 2 cm gradation marks. After a two-minute acclimation period, the vial was tapped three times and the ability to climb was video recorded. The percentage of flies able to climb 8 cm from the bottom of the vial within 20 s was calculated. A minimum of three separate trials were run for each group, and the mean (average climbing pass rate) was plotted as a single data point (Figs. 2g and 4c). Overall, 50 individuals or greater were analysed at each timepoint for every genotype.

**Longevity.** Supplementary Fig. 3a: Per data point, 8–10 male UAS > GFP/+; Hb9-Gal4/$W^{1118}$ flies were sorted upon eclosion and were maintained at 18, 25, or 29 °C on a 12-h light/dark cycle, at constant 50% humidity, transferred to fresh media every 2 days. Expired animals were counted and removed upon media transfer. Figures 1–4 and Supplementary Figs. 1–7: Per data point, 8–10 male flies of appropriate genotypes were sorted upon eclosion and were aged at 29 °C on a 12-h light/dark cycle, at constant 50% humidity and transferred to fresh media every 2 days. Expired animals were counted and removed upon media transfer.

**Measurement of glutamate receptor puncta size, active zone size and intensity.** To measure the size of glutamate receptor (GluRIID) and active zones (Bruchpilot) puncta, images of $mvim^T$ terminals were taken on a Zeiss 700 confocal microscope (×100 magnification). To measure the intensity of Brp protein, 3D Images with the step size of 350 nm were taken on an Inverted Olympus IX 83 motorized CSU W1 Spinning Disk microscope (×40 magnification) by using a UPlan S Apo 60x/1.42 oil immersion lens with a Orca Flash 4.0 camera. Presynaptic terminals of $mvim^T$ neurons were captured with a 488 nm Solid-state laser with BP 525/50 filter and the Brp signal was acquired with a 561 nm Pumped Diode Laser with a BP 605/70t filter in sequential mode.

Images were then analysed using Imaris (Oxford Instruments) using a custom script[79] (employing EasyXT from within Fiji). First, Presynaptic terminals were detected using a Surfaces Detector with the following parameters:

(1) Smoothing: 0.3 μm
(2) Local contrast filter: 3 μm
(3) Threshold: 50
(4) Filter: "Volume" between 1.00 and 3.000 μm³

Then individual active zones (Brp) or glutamate receptor puncta (GluRIID) were detected using a Spots Detector with the following parameters:

(1) Spot initial diameter: 0.310 μm (XY) and 0.620 μm (Z)
(2) Using local threshold and background subtraction
(3) Region diameter set from border
(4) Region local threshold: 0.6
(5) Filter: "Quality" above 50

Custom macros employed for these quantifications available to download[79].

**Statistical analysis and reproducibility.** Statistical analyses were performed in Prism v.7.0 (GraphPad Software). For Figs. 1, 2 and Supplementary Figs. 1–3, statistical significance for morphological and electrophysiological data were determined by Ordinary one-way ANOVA, followed by a Tukey post hoc test when multiple comparisons were required. A general linear model was utilized to analyse age-linked trends in ageing datasets (OriginLab, Massachusetts). For Figs. 3, 4 and Supplementary Figs. 4–7 electrophysiological experiments that included two groups, the results were analysed by Unpaired t-tests. Morphological analysis of NMJ synapses was conducted using Ordinary two-way ANOVA, and multiple comparisons were analysed by a Sidak's post hoc test. Sample sized were predetermined based on previous studies in the field (at least 50 for behavioral experiments and at least 5 for confocal experiments). All sample sizes are indicated in each figure legend and in the Supplementary Tables. 'n' values in text, figure legends and tables always represent biologically independent samples.

**Ethics statement.** This study employed *Drosophila melanogaster* which is not subject to institutional ethical approval. All relevant regulations for the use of genetically modified *Drosophila* were complied with.

**Reporting summary**. Further information on research design is available in the Nature Research Reporting Summary linked to this article.

## Data availability

Source data are provided as a Source Data file. Original microscopy images and electrophysiology datasets are available for download at https://doi.org/10.5281/zenodo.4906299[80]. Custom macros developed for this study are available for download at https://doi.org/10.5281/zenodo.4912667[79]. All other reagents including Drosophila strains are available upon request. Source data are provided with this paper.

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

## Acknowledgements
This work was supported in part using the resources and services of the BioImaging & Optics Platform (BIOP) Research Core Facility at the School of Life Sciences of EPFL. We are especially thankful for the assistance of Arne Seitz and Thierry Laroche. We are also very thankful to Robin Hiesinger and Troy Littleton for stocks and reagents, in addition to Hermann Aberle, Stephan Sigrist, Richard Mann and Joyce Fernandes. Stocks obtained from the Bloomington Drosophila Stock Center (NIH P40OD018537) and Vienna Drosophila Resource Center (VDRC) were used in this study. This work was supported by the Swiss National Science Foundation grant number: 31003A_179587 to B.M.

## Author contributions
S.B, S.V, W.J, B.C, E.R, J.A, O.B, S.S and B.M conceived and designed the study. S.B performed the majority of experiments and analysed the data, assisted by S.V. The electrophysiology preparation was formulated, experiments performed and analysed by S.V. Electron microscopy was performed by W.J. B.C. contributed preliminary data and insights. J.A. generated the novel GluRIIA allele. E.R. supervised the generation of antibodies and generated illustrations. O.B. aided the design and analysis of imaging experiments. S.S. generated the conditional *vglut* allele. S.B, S.V and B.M wrote the manuscript.

## Competing interests
The authors declare no competing interests.
