## [Peer Review File · Nature Communications]

REVIEWER COMMENTS

Reviewer #1 (Remarks to the Author):

In this paper, Banerjee et al. use adult *Drosophila* NMJ synapses to show that age-induced alterations in presynaptic structure is recapitulated by specifically reducing the frequency of spontaneous release. Furthermore, they show that increasing mini events in the motoneurons of aged animals rescues their presynaptic structure and motor ability. Overall, this in my opinion is an interesting manuscript that per se might well deserve publication in a high ranking journal. However, it would like to see their reaction concerning a couple of points:

- Fig1. Statistical analysis of differences in Fig 1 p, q, x might aid in appreciating the change in synaptic structure with aging. The synaptic changes aren't seen until day 60 whereas motor ability declines already at 20 days and drastically thereafter.
- Fig2. vGlut-/- animals continue to show deterioration with age, suggesting that mechanisms other than vesicular release are still at play in maintaining these synaptic structures. It would also be informative to test later time points (eg. 60 days where WT show phenotype) to observe progressive changes.
- The changes in synaptic structure in V100WFI animals compared to V100WT are convincing in showing that low mEJC frequency correlates with changes in synaptic structure, particularly combined with rescue with mCpXHB in fig3. However, evoked amplitudes still seem to be slightly affected 100WFI animals and cannot be completely ruled out.
- Furthermore, especially in the case of paraKD, it would be particularly informative to compare 60day old animals to ensure that they are not worse-off than WT animals.
- Fig 3. These results along with fig 2 are convincing as mentioned before. Are the mini frequencies affected throughout life in mCpxHB flies?
- Fig S3d y axis not consistent with other graphs
- The manuscript is clear and well written with only minor issues regarding missing words (eg. line 3).
- Fig 1SC total AZ numbers peak at 30 days! I do think that they should explicitly mention this in the main text. The finding is interesting also in the light of age-induced changes in the size of individual active zones (see Gupta et al., PlosBio 2016, PMID 27684064).
- I was wondering when comparing their results to the results of Mahoney, Rawson and Eaton, JN 2014, PMID: 24501352). They find that the amplitude of the EPSP abruptly INcreases during middle age and that this enhanced EPSP is maintained into late life, Important to calculate quantal contents. I am aware that we are talking different NMJs here, but still. Any way to reconcile these findings? Knowing how much aging phenotypes can depend on genetic background and diet/rearing conditions, I am kind of close asking them to visit the same NMJs as in this study in order to see...as this would be of importance for the field.
- I was also surprised to see that rescue from within motoneurons sufficed to rescue of climbing ability. I would have assumed that restoration of upstream circuits would be a necessity as well here..maybe they comment on this.
- Discussion for my taste is too much about their impression that minis have been neglected. I would rather like to hear more about their really functional role in maintenance, particularly if they say that evoked could not take over the job of minis.
- Would there be rescue by evoked only in absence of minis? Wonder what is so special about minis that evoked activation of glutamate receptors should not be able doing it..

Reviewer #2 (Remarks to the Author):

This brief and concise manuscript characterizes age-dependent changes in synaptic structure and function at the adult fly NMJ. The authors perform their studies using an adult NMJ that is well suited for the ageing processes and for the genetic manipulations they exploit to dissect the role of evoked and spontaneous neurotransmission. First, the authors show that in wild type NMJs, the boutons “fragment” and reduce in area along with a decrease in the density of release sites per bouton. This is accompanied by a reduction in evoked transmission, mini frequency, and climbing ability. Next, the authors employ several clever manipulations to show that this process is accelerated when no glutamate is released at NMJs, and when NMJs have evoked but not spontaneous neurotransmission. However, NMJs with miniature activity only do not seem to change the rate of NMJ fragmentation. Finally, the authors find that when miniature frequency is enhanced, age-dependent NMJ fragmentation is delayed and climbing ability may also improve. The authors conclude that spontaneous release (and really, it seems to be the rate of spontaneous release) is uniquely required for age-related changes at NMJs.

Overall this is an interesting story and adds the maintenance of NMJ integrity to a growing list of functions mediated by spontaneous neurotransmission. The authors deserve commendation for developing a series of clever genetic manipulations that enable the relatively clean uncoupling of evoked and spontaneous transmission. The manuscript is written with almost too much concision, with important details of how experiments were done kept out, but the overall take home message is exciting. Few neuroscientists currently believe miniature neurotransmission to be an “epiphenomenon” as stated in the abstract, but the results presented here not only provide evidence for another process in which minis are important, but also establishes a new genetic platform to study evoked vs miniature transmission in more detail. There are, however, several areas that the manuscript can be improved to both strengthen certain ideas and expand our understanding of the role of miniature transmission in maintaining NMJ integrity during the ageing process.

Major points:

1. Connection between single action potential evoked NMJ activity and ageing: The authors are to be commended for developing a series of manipulations that enable the relatively clear separation of evoked vs spontaneous neurotransmission. However, it seems there is room for other possible interpretations rather than the ones asserted in this manuscript. For example, as the authors know, NMJ activity in vivo is driven by tonic and phasic patterns of activity from two distinct motor neurons. This activity is quite different from the single action potential evoked release used to define the manipulations presented here. Perhaps alterations in short term plasticity over trains of high frequency stimulation in V100.WFI and/or Complexin.HB may actually contribute to some of the processes being studied. The authors should record short term trains of tonic/phasic activity in the V100 and Cpx NMJs to determine if patterns of evoked release are similarly not impacted compared to controls in addition to single action potential evoked release.

Differences in tonic vs phasic motor inputs: As the authors know, most NMJs in *Drosophila* are innervated by so called Type Is (phasic) and Ib (tonic) motor neurons which have distinct patterns of activity as well as major differences in synaptic structure. It is not clear from the current manuscript whether only type Ib NMJs were studied here, or both Is and Ib. Does the NMJ fragmentation the authors describe here happen similarly between Is and Ib NMJs? Are there differences?

2. Analysis of NMJ structure during ageing: The authors present “Bouton diameter” and “% single active zone boutons” as the primary descriptions of NMJ fragmentation over age in Fig. 1p,q (and subsequent figures). It is not clear why this somewhat arbitrary parameter is reported, rather than a more cumulative distribution of total bouton and active zone number per NMJ, as well as active zone size and/or intensity. Several papers have shown that the size/intensity and number of active zones can vary at fly NMJs without changes overall synaptic strength. This is illustrated in *rab3* mutants (Graf et al., *Neuron* 2009) and perhaps most dramatically in endophilin mutants (Goel et al., *JCB* 2019). The authors should present this more holistic data in the primary figures unless there is something particularly unique about boutons with single active zones. It seems that the more important analysis to know is 1) Total bouton number per NMJ; 2) Total active zone number per NMJ; and 3) Active zone size/intensity as NMJs age. It is noted that some of this information is available in supplementary data, but the overall trend is not clear.

This raises a larger question – what exactly do the authors think is happening to NMJs during ageing? They describe it as “fragmentation”, which implies that larger boutons are being reduced to smaller units, perhaps keeping the overall number of synapses (Brp and glutamate receptor pairs) similar. This is not the same as “degradation” or “degeneration”, which involves the loss of synapses and is the more conventional notion of ageing NMJs. Obviously it would be great to “watch” this process using live imaging to better understand what is happening, but perhaps that authors can explain in more detail exactly what they think is happening at fly NMJs as they age and how this compares to mammalian studies.

3. NMJ electrophysiology: The authors show a clear decrease in EJC amplitude and mEJC frequency through ageing, while no apparent changes in mini amplitude are observed. This raises several questions. If the total number of synapses/active zones remains the same at aged NMJs, then there must be a decrease in release probability per active zone, although the reason for reduced mini frequency is not clear if there is no change in the total number of release sites. In addition, from the supplemental images, it appeared that postsynaptic glutamate receptor fields were decreased in number or density but increased in size, is this correct? However, no change in mini amplitude suggests no change postsynaptic receptor levels. Perhaps the authors can explain in more detail how the synaptic electrophysiology is connected to the fragmentation in synaptic structure.

4. Another approach to test “mini only” NMJs: The authors use a clever Para-RNAi approach to generate an NMJ that expresses only mini release (and apparently no action potential evoked transmission). While convincing, it seems there are a few additional experiments that would serve to strengthen this data. First, the authors should record from NMJs across similar time points to ensure that the Para knock down is maintained to silence all evoked neurotransmission. From what I can tell, the electrophysiology is shown only at the early time point and does not demonstrate that evoked release remains blocked across all time points.

Second, and more importantly, the more conventional way to induce “mini only” transmission at the fly NMJ is to express tetanus toxin. This was the approach the authors used in previous publications (i.e. Choi et al., 2014, etc). The authors should validate their results observed using para-RNAi by expressing TNT, which should lead to similar effects as para-RNAi if mini release only is capable maintaining NMJ integrity.

5. Consistent conditions: In Fig 1, conventional culture temperature is used to show the age-

dependent NMJ fragmentation the authors are studying. However, due to the temperature-sensitive genetic manipulations the authors use in Figures 2 and 3, the authors apparently raise all their stocks, including controls, at 29C. For consistency and to more easily compare data, the authors should perform all their analyses in Figures 1-3 in the same conditions (29C), and show the conventional (and reduced temp data) in the supplement.

Furthermore, the authors should show their analyses at the same time points in all figures. Flies raised at 29C survive for ~40 days, but while NMJs are described through day 75 in Figure 1 as they are raised at 25, only values through day 20 are shown in Figure 2, while values through day 40 are shown in Figure 3. The authors should therefore present values through day 40 for flies raised at 29C in Figures 1-3.

6. Mechanism through which miniature activity stabilizes NMJs through age: Although the mechanism through which miniature neurotransmission stabilizes NMJ structure/function through the ageing process remains a mystery, it presumably involves a signaling cascade supported by miniature transmission and perhaps retrograde signaling. One obvious pathway well studied at the fly NMJ that also requires miniature activity and retrograde signaling is presynaptic homeostatic potentiation (PHP). It would be of interest to rule in or out PHP as being involved in this ageing process. One relatively easy approach would be to activate PHP signaling at NMJs and determine if age-dependent fragmentation occurs similarly to controls. This would be achieved by looking at GluRIIA mutants and/or activating PHP by muscle overexpression of Target of Rapamycin (TOR; see Penney et al., *Neuron* 2012; Goel et al., *Cell Reports*, 2017). In addition, the authors could examine mutants in which PHP is blocked (i.e., dysbindin mutants, etc).

Minor comments:

1. It is not at all clear what the genotypes are for the NMJ stainings presented in Fig. 1. In some panels GFP is presumably being driven in motor neurons, in others mCherry is being driven, and in others peripheral glia are being labeled by mCherry. The authors need to be very specific about what the genotypes are for each of as well as the controls in the Figure legend. Is HB9-Gal4 driving UAS-GFP? Similarly with mCherry labeled NMJs – what is the genotype?

2. The “set point” of synaptic strength has been reported to change at adult fly NMJs as a function of ageing (Mahoney R et al., *J. Neurosci* 2014). Have the authors examined this at the NMJs that express HB9? A dynamic change in the set point of synaptic strength through ageing would obviously complicate this study. At minimum, the authors should consider this possibility and perform appropriate controls.

- Signed Dion Dickman

RESPONSE TO REVIEWERS

Reviewer #1

In this paper, Banerjee et al. use adult *Drosophila* NMJ synapses to show that age-induced alterations in presynaptic structure is recapitulated by specifically reducing the frequency of spontaneous release. Furthermore, they show that increasing mini events in the motoneurons of aged animals rescues their presynaptic structure and motor ability.

Overall, this in my opinion is an interesting manuscript that per se might well deserve publication in a high ranking journal. However, it would like to see their reaction concerning a couple of points:

- Fig1. Statistical analysis of differences in Fig 1 p, q, x might aid in appreciating the change in synaptic structure with aging. The synaptic changes aren't seen until day 60 whereas motor ability declines already at 20 days and drastically thereafter.

We thank the reviewer for their supportive comments. To address this issue, we have added a new figure (Fig. 2g) which allows a direct comparison of key ageing parameters including bouton fragmentation, EJC amplitude, mEJC frequency and motor ability. The legend for this figure describes the timepoint at which each parameter becomes significantly different from young animals and full statistics of all timepoints are also included in Supplementary Table 2.

- Fig2. vGlut^{-/-} animals continue to show deterioration with age, suggesting that mechanisms other than vesicular release are still at play in maintaining these synaptic structures. It would also be informative to test later time points (eg. 60 days where WT show phenotype) to observe progressive changes.

The reviewer raises an interesting point. We have carried out the analysis they suggested by examining animals continuously until 35 days at 29°C (equivalent to 60 days at 25°C (Supplementary Fig. 3b, 29°C is necessary for tsGAL80 induction for Vglut removal). We found that both bouton diameter, active zone number and the number of single active zone boutons were not different between Vglut mutants with continued ageing after a maximal amount was achieved (49% reduction in bouton size, 81% increase the number of single active zones) see Fig. 3b. This data suggests that bouton fragmentation rapidly reaches a 'ceiling' when vglut is removed with no additional fragmentation occurring in response to further ageing. This result together with V100^{WFI} and mCpx^{HB} is consistent with a decline in vesicular release being the core mechanism underlying age-dependent bouton fragmentation. We have added additional text and discussion of this experiment.

- The changes in synaptic structure in V100WFI animals compared to V100WT are convincing in showing that low mEJC frequency correlates with changes in synaptic structure, particularly combined with rescue with mCpXHB in fig3. However, evoked amplitudes still seem to be slightly affected 100WFI animals and cannot be completely ruled out.

The point that the reviewer makes is well taken and similar issues were raised by reviewer #2. To address if differences in evoked release could be contributing to the phenotypes we observed, we eliminated evoked release in V100^{WFI} mutant animals by inhibition of Para and examined if morphological parameters were additionally altered compared to V100^{WFI} mutants where evoked release was intact. As shown in Fig.3e and Supplementary Fig.5g,h we observed no differences in bouton fragmentation when evoked release was present or absent

in V100^{WFI} mutants. Similar results were found for Complexin (Supplementary Fig. 7b,f). This is consistent with miniature neurotransmission being singularly required to maintain synaptic structures.

Furthermore, especially in the case of paraKD, it would be particularly informative to compare 60day old animals to ensure that they are not worse-off than WT animals.

We have carried out the analysis the reviewer suggested by examining animals continuously aged until 35 days at 29°C [equivalent to 60 days at 25°C, Supplementary Fig. 3b, 29°C is necessary for tsGAL80 adult induction of Para RNAi expression]. As shown in Fig. 3d, we see no differences in terminal morphology throughout lifespan between Para inhibited animals and controls (confirmatory electrophysiology Supplementary Fig. 4i). This is consistent with loss of evoked neurotransmission not contributing to terminal structural dissolution.

Fig 3. These results along with fig 2 are convincing as mentioned before. Are the mini frequencies affected throughout life in mCpxHB flies?

We have carried out the analysis the reviewer suggested by examining animals continuously aged again until 35 days at 29°C. We observed that mEJC frequency remained elevated until old age in mCpxHB animals. We have added this result to the text and this data to Supplementary Table 6.

- Fig S3d y axis not consistent with other graphs

Thanks for pointing this out. We have rectified this error.

The manuscript is clear and well written with only minor issues regarding missing words (eg. line 3).

Thanks for pointing this out. We have proofread the manuscript again.

- Fig 1SC total AZ numbers peak at 30 days! I do think that they should explicitly mention this in the main text. The finding is interesting also in the light of age-induced changes in the size of individual active zones (see Gupta et al., PlosBio 2016, PMID 27684064).

We agree with the reviewer this is an important result especially in light of prior studies in the CNS. To highlight this link, we have now moved this data to Fig.1q and explicitly make the link to Gupta et al. 2016 in the text.

- I was wondering when comparing their results to the results of Mahoney, Rawson and Eaton, JN 2014, PMID: 24501352). They find that the amplitude of the EPSP abruptly INcreases during middle age and that this enhanced EPSP is maintained into late life, Important to calculate quantal contents. I am aware that we are talking different NMJs here, but still. Any way to reconcile these findings? Knowing how much aging phenotypes can depend on genetic background and diet/rearing conditions, I am kind of close asking them to visit the same NMJs as in this study in order to see...as this would be of importance for the field.

We agree that our results do not easily reconcile with those of Mahoney et al., 2014¹. We would first like to point out that we believe our methods are technically superior. For this manuscript and preparation, we developed a discontinuous voltage clamp method. We did this as current recordings, as in Mahoney et al.,2014 may be susceptible to unclamped voltage sensitive currents and/or subject to covariate resting membrane potential (RMP) alterations

particularly during ageing, which are alleviated in our study through recording of voltage clamped synaptic currents. Moreover, we have now confirmed our results by making complementary measurements of the readily releasable pool (RRP) of synaptic vesicles in young and old terminals (Supplementary Fig. 2e). Consistent with our EJC and quantal content measurements, we observe a 56% decline of the RRP in young versus old animals.

Increased quantal content during ageing had been shown in mammalian ageing models^{2,3} however later studies have refuted this^{4,5}. Even given this debate, the mammalian data are significantly different to the findings of Mahoney et al. Mammalian NMJs display confounding and dramatic reductions in quantal size and a hyperpolarised RMP together with a large increase in animal weight². Mahoney et al. report no alteration in quantal size but increased EJP amplitudes alone.

Other studies in addition to ours rather support a decline in evoked amplitudes during ageing. Martinez et al.⁶ examined ageing of the *Drosophila* Giant Fibre system and observed a progressive decline of DLM and TTM in response to high frequency stimulation with ageing. Liu et al.⁷ also observed a decline in evoked cholinergic synaptic currents together with a reduction in miniature frequency during ageing in *C.elegans*. Therefore, we believe our results are consistent with the majority of NMJ ageing studies though we cannot exclude differences may exist in the proboscis terminals studied by Mahoney et al. We have noted this differences in the revised discussion.

• I was also surprised to see that rescue from within motoneurons sufficed to rescue of climbing ability. I would have assumed that restoration of upstream circuits would be a necessity as well here..maybe they comment on this.

This result was also surprising to us. We speculate that improving the terminal structure may preserve muscle function and/or proprioceptive feedback perhaps through transsynaptic trophic factors. Of note, preserving NMJ structures alone can prolong motor activity in rodent models of ALS⁸.

Discussion for my taste is too much about their impression that minis have been neglected. I would rather like to hear more about their really functional role in maintenance, particularly if they say that evoked could not take over the job of minis. Would there by rescue by evoked only in absence of minis? Wonder what is so special about minis that evoked activation of glutamate receptors should not be able doing it..

As the reviewer will appreciate, we and others^{9,10} are intensively investigating mechanisms by which miniature events can be discriminated from evoked release. This study and our previous work on the postsynaptic manipulation of miniature neurotransmission in bouton development⁹ support miniature events have singular functions and therefore this discrimination must occur. We have expanded our discussion to speculate upon possible mechanisms by which this might be achieved.

Reviewer #2

This brief and concise manuscript characterizes age-dependent changes in synaptic structure and function at the adult fly NMJ. The authors perform their studies using an adult NMJ that is well suited for the ageing processes and for the genetic manipulations they exploit to dissect the role of evoked and spontaneous neurotransmission. First, the authors show that in wild type NMJs, the boutons “fragment” and reduce in area along with a decrease in the density of release sites per bouton. This is accompanied by a reduction in evoked transmission, mini frequency, and climbing ability. Next, the authors employ several clever manipulations to show that this process is accelerated when no glutamate is released at NMJs, and when NMJs have evoked but not spontaneous neurotransmission. However, NMJs with miniature activity only do not seem to change the rate of NMJ fragmentation. Finally, the authors find that when miniature frequency is enhanced, age-dependent NMJ fragmentation is delayed and climbing ability may also improve. The authors conclude that spontaneous release (and really, it seems to be the rate of spontaneous release) is uniquely required for age-related changes at NMJs.

Overall this is an interesting story and adds the maintenance of NMJ integrity to a growing list of functions mediated by spontaneous neurotransmission. The authors deserve commendation for developing a series of clever genetic manipulations that enable the relatively clean uncoupling of evoked and spontaneous transmission. The manuscript is written with almost too much concision, with important details of how experiments were done kept out, but the overall take home message is exciting. Few neuroscientists currently believe miniature neurotransmission to be an “epiphenomenon” as stated in the abstract, but the results presented here not only provide evidence for another process in which minis are important, but also establishes a new genetic platform to study evoked vs miniature transmission in more detail. There are, however, several areas that the manuscript can be improved to both strengthen certain ideas and expand our understanding of the role of miniature transmission in maintaining NMJ integrity during the ageing process.

Major points:

1. Connection between single action potential evoked NMJ activity and ageing: The authors are to be commended for developing a series of manipulations that enable the relatively clear separation of evoked vs spontaneous neurotransmission. However, it seems there is room for other possible interpretations rather than the ones asserted in this manuscript. For example, as the authors know, NMJ activity in vivo is driven by tonic and phasic patterns of activity from two distinct motor neurons. This activity is quite different from the single action potential evoked release used to define the manipulations presented here. Perhaps alterations in short term plasticity over trains of high frequency stimulation in V100.WFI and/or Complexin.HB may actually contribute to some of the processes being studied. The authors should record short term trains of tonic/phasic activity in the V100 and Cpx NMJs to determine if patterns of evoked release are similarly not impacted compared to controls in addition to single action potential evoked release.

We thank Prof. Dickman for his supportive comments and careful evaluation of the work. The point that the reviewer makes is valid and similar issues were raised by reviewer #1. There are indeed a large number of possible evoked release parameters that we could conceivably measure in V100^{WFI} and Cpx^{HB} mutants but we would still be left with the issue to establish (or not) the relationship of any potential changes to the morphological and behavioral changes we observe. Therefore, we decided to simply eliminate evoked release entirely in V100 and Cpx mutants by co-inhibiting Para and evaluate if this induced any additional changes to bouton morphology. As shown in Supplementary Fig. 5g,h and Supplementary Fig. 7b,f we

observed no differences in bouton fragmentation when evoked release was present or absent in V100 or Cpx mutants. These results are consistent with miniature neurotransmission being singularly required to maintain synaptic structures independently of evoked release. We have expanded the text to discuss these results.

Differences in tonic vs phasic motor inputs: As the authors know, most NMJs in *Drosophila* are innervated by so called Type Is (phasic) and Ib (tonic) motor neurons which have distinct patterns of activity as well as major differences in synaptic structure. It is not clear from the current manuscript whether only type Ib NMJs were studied here, or both Is and Ib. Does the NMJ fragmentation the authors describe here happen similarly between Is and Ib NMJs? Are there differences?

The presence and relationship between tonic and phasic terminals on these muscle fibers had not previously been described so at the suggestion of the reviewer we have undertaken this task. We generated a monoclonal antibody against *Drosophila* VMAT which allowed us to discriminate between phasic neuropeptide co-releasing boutons and tonic VGLUT only releasing boutons. Importantly, HB9>Gal4, used for the majority of manipulations in this manuscript, is only expressed in neurons producing the tonic terminal upon these muscles (mvim^T) but not the phasic terminal (mvim^P) (Supplementary Fig. 1a). Unlike many larval muscles, boutons from tonic and phasic terminals are tightly intercalated on these adult muscles (Supplementary Fig. 1a). Phasic terminal (VMAT labeled boutons) are much smaller on average ($0.69\pm 0.02\mu\text{m}$) than tonic boutons ($1.57\pm 0.04\mu\text{m}$) and predominantly have only one active zone. Given these factors, it was not possible to examine fragmentation of these phasic boutons.

2. Analysis of NMJ structure during ageing: The authors present “Bouton diameter” and “% single active zone boutons” as the primary descriptions of NMJ fragmentation over age in Fig. 1p,q (and subsequent figures). It is not clear why this somewhat arbitrary parameter is reported, rather than a more cumulative distribution of total bouton and active zone number per NMJ, as well as active zone size and/or intensity. Several papers have shown that the size/intensity and number of active zones can vary at fly NMJs without changes overall synaptic strength. This is illustrated in rab3 mutants (Graf et al., Neuron 2009) and perhaps most dramatically in endophilin mutants (Goel et al., JCB 2019). The authors should present this more holistic data in the primary figures unless there is something particularly unique about boutons with single active zones. It seems that the more important analysis to know is 1) Total bouton number per NMJ; 2) Total active zone number per NMJ; and 3) Active zone size/intensity as NMJs age. It is noted that some of this information is available in supplementary data, but the overall trend is not clear.

As the reviewer noted, some of this data was available in tables in previous manuscript version, however we have now added additional figures and performed new experiments to address the reviewer’s concern. Bouton number data has been added as Supplementary Fig. 1d. Consistent with larger young boutons subdividing into multiple smaller boutons in old animals, the number of bouton varicosities increases with age. We felt it better to add this figure to supplementary data to avoid the potential for confusion to the casual reader that the terminal size is increasing (as similar data in larval terminals for example could suggest) rather than in fact that boutons are fragmenting and that total terminal size is decreasing (Fig1p). We have added the total active number per terminal now to Fig. 1q as was also suggested by reviewer #1. As requested, we have now measured active zone size and Brp intensity during ageing (Supplementary Fig. 2a,b). We found that active zone size decreases with ageing while active zone intensity increases (consistent with CNS measurements by Gupta et al. 2016). It

is important to note that BRP manipulation does not alter miniature neurotransmission frequency¹¹. We have noted these additions in the revised text.

This raises a larger question – what exactly do the authors think is happening to NMJs during ageing? They describe it as “fragmentation”, which implies that larger boutons are being reduced to smaller units, perhaps keeping the overall number of synapses (Brp and glutamate receptor pairs) similar. This is not the same as “degradation” or “degeneration”, which involves the loss of synapses and is the more conventional notion of ageing NMJs. Obviously it would be great to “watch” this process using live imaging to better understand what is happening, but perhaps that authors can explain in more detail exactly what they think is happening at fly NMJs as they age and how this compares to mammalian studies.

The scenario the reviewer outlines is exactly what we think occurs and is supported by our data. To illustrate our model, we have added a new diagram (Fig. 5). Larger multiple active zone boutons in young animals subdivide into progressively smaller and ultimately single active zone boutons as ageing progresses. As we can only reliably measure the most extreme extent of this subdivision (i.e. to single active zone boutons) we are likely underestimating the total extent of fragmentation, but as the reviewer points out, without live imaging during ageing (not currently possible), we cannot be certain of the prior number of active zones in any individual bouton. Nonetheless, the cumulative increase in the number of single active zone boutons, decrease in bouton diameter, increase in the total number of (albeit smaller) boutons, presence of multiple presynaptic boutons opposed to a single postsynaptic elaboration in old animals (in contrast to young animals) are all consistent with fragmentation of boutons as ageing progresses. Of note, related phenomena have been reported at both mammalian and human neuromuscular junctions and we have expanded upon our discussion including additional references to support this comparison.

3. NMJ electrophysiology: The authors show a clear decrease in EJC amplitude and mEJC frequency through ageing, while no apparent changes in mini amplitude are observed. This raises several questions. If the total number of synapses/active zones remains the same at aged NMJs, then there must be a decrease in release probability per active zone, although the reason for reduced mini frequency is not clear if there is no change in the total number of release sites. In addition, from the supplemental images, it appeared that postsynaptic glutamate receptor fields were decreased in number or density but increased in size, is this correct? However, no change in mini amplitude suggests no change postsynaptic receptor levels. Perhaps the authors can explain in more detail how the synaptic electrophysiology is connected to the fragmentation in synaptic structure.

The total number of active zones does not remain constant during ageing. As shown in Fig. 1q, the total number of active zones peak in middle age and then decline in older animals. As suggested by the reviewer, we have also now measured postsynaptic receptor puncta size (with GlurIID) during ageing (Supplementary Fig. 2d). We found a 21% ($P < 0.001$) increase in puncta size from young to middle aged flies and which remained constant from middle age to old age. In contrast, miniature event amplitudes remain constant during ageing (apart from the very oldest animals) and while both evoked amplitudes and miniature event frequency decline. Therefore, no obvious correlation exists between either postsynaptic glutamate receptor puncta size or presynaptic active zone number and mEJC frequency, consistent with previous results¹¹. We note that in larval synapses^{12,13}, individual active zones may have divergent miniature and evoked release properties. An interesting topic for future studies is to investigate if this segregation also occurs at adult terminals. We have added this additional data to the results and expanded the discussion.

4. Another approach to test “mini only” NMJs: The authors use a clever Para-RNAi approach to generate an NMJ that expresses only mini release (and apparently no action potential evoked transmission). While convincing, it seems there are a few additional experiments that would serve to strengthen this data. First, the authors should record from NMJs across similar time points to ensure that the Para knock down is maintained to silence all evoked neurotransmission. From what I can tell, the electrophysiology is shown only at the early time point and does not demonstrate that evoked release remains blocked across all time points.

As the reviewer requested, we have recorded from older animals where para is knocked down and can confirm that the evoked release remains inhibited (Supplementary Fig.4i and Table 6).

Second, and more importantly, the more conventional way to induce “mini only” transmission at the fly NMJ is to express tetanus toxin. This was the approach the authors used in previous publications (i.e. Choi et al., 2014, etc). The authors should validate their results observed using para-RNAi by expressing TNT, which should lead to similar effects as para-RNAi if mini release only is capable maintaining NMJ integrity.

As we⁹ and others¹⁵ have found, expression of tetanus toxin can partially inhibit miniature event frequency in embryonic neurons in addition to inhibiting evoked release. In larva, miniature neurotransmission subsequently recovers to normal levels⁹. We found that inducing expression of tetanus toxin in adult neurons also partially inhibited miniature event frequency in addition to blocking evoked release at least in younger animals. Therefore, to address the reviewer’s concern, we generated and expressed an enhanced version of Kir2.1(UAS>Kir_{2.1E}) in adult motor neurons which does not alter miniature event frequency (Supplementary Fig. 4j and Table 2). Similar to Para inhibition, Kir2.1 expression does not alter synaptic morphology (Supplementary Fig. 4k and Table 3). This result has been added to the text.

5. Consistent conditions: In Fig 1, conventional culture temperature is used to show the age-dependent NMJ fragmentation the authors are studying. However, due to the temperature-sensitive genetic manipulations the authors use in Figures 2 and 3, the authors apparently raise all their stocks, including controls, at 29C. For consistency and to more easily compare data, the authors should perform all their analyses in Figures 1-3 in the same conditions (29C), and show the conventional (and reduced temp data) in the supplement.

We understand the reviewer’s point, but we feel it important to first describe the observations we have made of ageing of this terminal at conventional culture temperatures in manuscript (25°C). To address their issue, we have now added a new Supplementary Fig. 3 which mirrors the data in Fig. 1 but entirely at 29°C culture temperatures.

Furthermore, the authors should show their analyses at the same time points in all figures. Flies raised at 29C survive for ~40 days, but while NMJs are described through day 75 in Figure 1 as they are raised at 25, only values through day 20 are shown in Figure 2, while values through day 40 are shown in Figure 3. The authors should therefore present values through day 40 for flies raised at 29C in Figures 1-3.

We have added additional time points as the reviewer suggested to Fig. 3.

6. Mechanism through which miniature activity stabilizes NMJs through age: Although the mechanism through which miniature neurotransmission stabilizes NMJ structure/function through the ageing process remains a mystery, it presumably involves a signaling cascade supported by miniature transmission and perhaps retrograde signaling. One obvious pathway

well studied at the fly NMJ that also requires miniature activity and retrograde signaling is presynaptic homeostatic potentiation (PHP). It would be of interest to rule in or out PHP as being involved in this ageing process. One relatively easy approach would be to activate PHP signaling at NMJs and determine if age-dependent fragmentation occurs similarly to controls. This would be achieved by looking at GluRIIA mutants and/or activating PHP by muscle overexpression of Target of Rapamycin (TOR; see Penney et al., *Neuron* 2012; Goel et al., *Cell Reports*, 2017). In addition, the authors could examine mutants in which PHP is blocked (i.e., dysbindin mutants, etc).

To address the reviewer's point, we examined adult mutants of GlurIIA. We found that PHP was induced in these animals similar to established mechanisms at larval terminals (Supplementary Fig. 6a). However, we observed no change in bouton morphology in GlurIIA mutants compared to controls (Supplementary Fig. 6b). We note the PHP also appears fully independent of the role for miniature neurotransmission in bouton development in larva⁹. We have noted the results of this experiment in the revised text.

Minor comments:

1. It is not at all clear what the genotypes are for the NMJ stainings presented in Fig. 1. In some panels GFP is presumably being driven in motor neurons, in others mCherry is being driven, and in others peripheral glia are being labeled by mCherry. The authors need to be very specific about what the genotypes are for each of as well as the controls in the Figure legend. Is HB9-Gal4 driving UAS-GFP? Similarly with mCherry labeled NMJs – what is the genotype?

We have added a table with full genotypes for these figures (Supplementary Table 1) in addition to clarifying where necessary in other figure legends.

2. The “set point” of synaptic strength has been reported to change at adult fly NMJs as a function of ageing (Mahoney R et al., *J. Neurosci* 2014). Have the authors examined this at the NMJs that express HB9? A dynamic change in the set point of synaptic strength through ageing would obviously complicate this study. At minimum, the authors should consider this possibility and perform appropriate controls.

Reviewer #1 also asked this question. In short, we do not observe any age-dependent increase in “set-point” or more precisely quantal content at this terminal (Supplementary Fig. 2e). Instead, we observe a continual decline of quantal content consistent with a progressive diminishment of evoked neurotransmission amplitudes while miniature event amplitudes remain constant. As described also for Reviewer #1, we further confirmed our results by making complementary measurements of the ready releasable pool (RRP) of synaptic vesicles in young and old terminals (Supplementary Fig. 2e). Consistent with our EJC and quantal content measurements, we observe a 56% decline of the RRP in young versus old animals. Our results at this terminal are similar to many other studies in *Drosophila*, *C. elegans* and mammalian terminals^{3,4,6,7,14}. We have expounded upon these contrasting results in the revised discussion.

References:

- 1 Mahoney, R. E., Rawson, J. M. & Eaton, B. A. An age-dependent change in the set point of synaptic homeostasis. *The Journal of neuroscience : the official journal of the Society for Neuroscience* **34**, 2111-2119, doi:10.1523/JNEUROSCI.3556-13.2014 (2014).
- 2 Kelly, S. S. The effect of age on neuromuscular transmission. *The Journal of physiology* **274**, 51-62, doi:10.1113/jphysiol.1978.sp012133 (1978).
- 3 Banker, B. Q., Kelly, S. S. & Robbins, N. Neuromuscular transmission and correlative morphology in young and old mice. *The Journal of physiology* **339**, 355-377, doi:10.1113/jphysiol.1983.sp014721 (1983).
- 4 Coggan, J. S. *et al.* Age-associated synapse elimination in mouse parasympathetic ganglia. *Journal of neurobiology* **60**, 214-226, doi:10.1002/neu.20022 (2004).
- 5 Barnes, C. A., Rao, G. & Orr, G. Age-related decrease in the Schaffer collateral-evoked EPSP in awake, freely behaving rats. *Neural plasticity* **7**, 167-178, doi:10.1155/NP.2000.167 (2000).
- 6 Martinez, V. G. *et al.* Age-related changes in climbing behavior and neural circuit physiology in *Drosophila*. *Developmental neurobiology* **67**, 778-791, doi:10.1002/dneu.20388 (2007).
- 7 Liu, J. *et al.* Functional aging in the nervous system contributes to age-dependent motor activity decline in *C. elegans*. *Cell metabolism* **18**, 392-402, doi:10.1016/j.cmet.2013.08.007 (2013).
- 8 Cantor, S. *et al.* Preserving neuromuscular synapses in ALS by stimulating MuSK with a therapeutic agonist antibody. *eLife* **7**, doi:10.7554/eLife.34375 (2018).
- 9 Choi, B. J. *et al.* Miniature neurotransmission regulates *Drosophila* synaptic structural maturation. *Neuron* **82**, 618-634, doi:10.1016/j.neuron.2014.03.012 (2014).
- 10 Andrae, L. C. & Burrone, J. The role of spontaneous neurotransmission in synapse and circuit development. *J Neurosci Res* **96**, 354-359, doi:10.1002/jnr.24154 (2018).
- 11 Wagh, D. A. *et al.* Bruchpilot, a protein with homology to ELKS/CAST, is required for structural integrity and function of synaptic active zones in *Drosophila*. *Neuron* **49**, 833-844, doi:10.1016/j.neuron.2006.02.008 (2006).
- 12 Peled, E. S., Newman, Z. L. & Isacoff, E. Y. Evoked and spontaneous transmission favored by distinct sets of synapses. *Current biology : CB* **24**, 484-493, doi:10.1016/j.cub.2014.01.022 (2014).
- 13 Melom, J. E., Akbergenova, Y., Gavornik, J. P. & Littleton, J. T. Spontaneous and evoked release are independently regulated at individual active zones. *The Journal of neuroscience : the official journal of the Society for Neuroscience* **33**, 17253-17263, doi:10.1523/JNEUROSCI.3334-13.2013 (2013).
- 14 Barnes, C. A. & McNaughton, B. L. Physiological compensation for loss of afferent synapses in rat hippocampal granule cells during senescence. *The Journal of physiology* **309**, 473-485, doi:10.1113/jphysiol.1980.sp013521 (1980).
15. Sweeney ST, Broadie K, Keane J, Niemann H, O'Kane CJ. Targeted expression of tetanus toxin light chain in *Drosophila* specifically eliminates synaptic transmission and causes behavioral defects. *Neuron*. 341-51. doi: 10.1016/0896-6273(95)90290-2 (1995).

REVIEWER COMMENTS

Reviewer #1 (Remarks to the Author):

I am satisfied with their revision and suggest acceptance as is.

Reviewer #2 (Remarks to the Author):

The authors have done an excellent job of responding to my central concerns with new controls, data, and analyses in this revised manuscript. In particular, the co-inhibition of para along with v100 and cpx manipulations was a good way to control for the impacts of residual evoked release. The work that went into evaluating phasic releasing NMJs in addition to the tonic HB9 positive inputs in this adult system is also appreciated. Finally, the inclusion of the Kir2.1 experiment to control for para RNAi is also a very nice additional experiment. Overall, this revised manuscript has been strengthened and the central findings extended and should be of interest and impact to a variety of researchers.

I do have one small minor comment for the authors. The dGluRIIA.JA1 allele – The description of generating these mutants is sufficiently detailed and the immunostaining-based screen to identify mutants is straightforward. However, the molecular lesion of this allele is not defined (assume it is a frameshift?). Did the authors not sequence this allele? It would be of interest to define the molecular change of this allele in the final version of the report.

RESPONSE TO REVIEWERS

Reviewer #1 (Remarks to the Author):

I am satisfied with their revision and suggest acceptance as is.

We are gratified to have satisfied the reviewer and thank them for their helpful insights.

Reviewer #2 (Remarks to the Author):

The authors have done an excellent job of responding to my central concerns with new controls, data, and analyses in this revised manuscript. In particular, the co-inhibition of para along with v100 and cpx manipulations was a good way to control for the impacts of residual evoked release. The work that went into evaluating phasic releasing NMJs in addition to the tonic HB9 positive inputs in this adult system is also appreciated. Finally, the inclusion of the Kir2.1 experiment to control for para RNAi is also a very nice additional experiment. Overall, this revised manuscript has been strengthened and the central findings extended and should be of interest and impact to a variety of researchers.

I do have one small minor comment for the authors. The dGluRIIA.JA1 allele – The description of generating these mutants is sufficiently detailed and the immunostaining-based screen to identify mutants is straightforward. However, the molecular lesion of this allele is not defined (assume it is a frameshift?). Did the authors not sequence this allele? It would be of interest to define the molecular change of this allele in the final version of the report.

We are thankful for the kind comments of the reviewer. As requested, we have added the exact molecular nature of the novel dGluRIIA allele JA1 (a three nucleotide deletion in the ligand binding domain encoding region) to the methods.